# Langerhans cells shape postnatal oral homeostasis in a mechanical-force-dependent but microbiota and IL17-independent manner

Yasmin Jaber[1], Yasmine Netanely[1], Reem Naamneh[1], Or Saar[2], Khaled Zubeidat[1], Yasmin Saba[1], Olga Georgiev[1], Paz Kles[2], Or Barel[1], Yael Horev[2], Omri Yosef [3], Luba Eli-Berchoer[1], Chen Nadler[4,5], Gili Betser-Cohen[6], Hagit Shapiro[7], Eran Elinav [7,8], Asaf Wilensky[2] & Avi-Hai Hovav [1] ✉

The postnatal interaction between microbiota and the immune system establishes lifelong homeostasis at mucosal epithelial barriers, however, the barrier-specific physiological activities that drive the equilibrium are hardly known. During weaning, the oral epithelium, which is monitored by Langerhans cells (LC), is challenged by the development of a microbial plaque and the initiation of masticatory forces capable of damaging the epithelium. Here we show that microbial colonization following birth facilitates the differentiation of oral LCs, setting the stage for the weaning period, in which adaptive immunity develops. Despite the presence of the challenging microbial plaque, LCs mainly respond to masticatory mechanical forces, inducing adaptive immunity, to maintain epithelial integrity that is also associated with naturally occurring alveolar bone loss. Mechanistically, masticatory forces induce the migration of LCs to the lymph nodes, and in return, LCs support the development of immunity to maintain epithelial integrity in a microbiota-independent manner. Unlike in adult life, this bone loss is IL-17-independent, suggesting that the establishment of oral mucosal homeostasis after birth and its maintenance in adult life involve distinct mechanisms.

The mutual development of the immune system and the microbiota after birth are necessary to reach homeostasis in mucosal barriers, also capable of influencing long-term host-microbiota relationships that might have health implications in adult life[1]. Nevertheless, postnatal development often includes barrier-specific physiological processes that are required for the proper functioning of the organ. Whether such unique developmental processes are also involved in the initial establishment of homeostasis in certain mucosal barriers requires further clarification.

[1]Institute of Biomedical and Oral Research, Faculty of Dental Medicine, Hebrew University, Jerusalem, Israel. [2]Faculty of Dental Medicine, Hebrew University, Jerusalem, Israel; Department of Periodontology, Hadassah Medical Center, Jerusalem, Israel. [3]The Lautenberg Center for Immunology and Cancer Research, Israel-Canada Medical Research Institute, Faculty of Medicine, Hebrew University, Jerusalem, Israel. [4]Faculty of Dental Medicine, Hebrew University, Jerusalem, Israel. [5]Department of Oral Medicine, Sedation & Maxillofacial Imaging, Hadassah Medical Center, Jerusalem, Israel. [6]Division of Identification and Forensic Science, Police National HQ, Jerusalem, Israel. [7]System Immunology Department, Weizmann Institute of Science, Rehovot, Israel. [8]Microbe & Cancer Division, DKFZ, Heidelberg, Germany. ✉e-mail: avihaih@ekmd.huji.ac.il

The early-life crosstalk between the oral immune system and the microbiota has recently begun to be discovered, demonstrating complex and unique mechanisms to the oral mucosa[2,3]. Early after birth, the oral mucosa is colonized with an immense amount of microbiota, a period when the epithelium is permeable and immunologically immature. Neutrophils are recruited by IL-17-producing γδT cells to protect the vulnerable epithelium, disappearing until weaning as the epithelium is sealed and acquires adult immunological functions in a microbiota-dependent fashion. Simultaneously, the parotid salivary glands complete their development until weaning, secreting optimal saliva flow with antimicrobial activity that sharply reduces the microbial load to adult levels[2]. Yet, the postnatal development of the oral mucosa includes also the eruption of the teeth and the formation of the gingiva, enabling the transition from suckling to a solid diet, a period termed weaning. This initiates mastication that exerts mechanical forces damaging the gingival epithelium, representing a unique immunological challenge to the oral immune system. In a seminal study, Dutzan *et al.* have shown that throughout life, epithelial damage caused by ongoing mastication drives a local proliferation of IL-17-producing CD4+ T (Th17) cells[4]. These Th17 cells play a tissue-protective role at a steady state but can also mediate alveolar bone loss under inflammatory conditions. While this mastication-Th17 axis was reported as age-dependent[4], whether and how early-life mastication regulates the initial establishment of gingival immunity after birth remains unclear. This is of great importance because the immune response established in the gingiva postnatally may affect the susceptibility to develop periodontitis in adult life, a common oral disease with systemic clinical consequences[5].

As the antigen-presenting cells (APC) of stratified epithelia, Langerhans cells (LC) represent key sentinels of the oral epithelium[6]. Unlike skin LCs, oral LCs arise from bone marrow precursors rather than embryonic precursors and receive differentiation instructions within the epithelium[7,8]. Skin LCs were shown to respond to a variety of immunological insults, such as microbial invasion and mechanical damage, capable of inducing immunity or tolerance in a challenge-dependent context[9]. With regards to oral LCs, their prolonged depletion in adult life dysregulates oral immunity and microbiota, resulting in accelerated bone-destructing immunity[7]. Moreover, conditional ablation of LCs during pathogen-induced periodontitis increases alveolar bone loss, the hallmark of this disease, due to reduced generation of T regulatory (Treg) cells[10]. Another study employing a murine model in which LCs are absent from birth reported that while LCs induce Th17 cells during experimental periodontitis, they have no impact on bone loss[11]. These findings demonstrate that LCs control oral mucosal immunity in adult life. However, it is not clear whether LCs are involved in the establishment of oral immunity after birth. Furthermore, whether LCs respond to the developing bacterial plaque on the teeth and/or to epithelial damage caused by masticatory forces early in life is unknown.

This work demonstrates that while the microbiota is necessary for the development of gingival LCs after birth, during weaning, LCs respond to the masticatory forces rather than the microbiota, shaping the establishment of gingival immunity. This LC-mediated immunity in early-life associates with alveolar bone loss, which, unlike in adult life, is not mediated by IL-17. This study reveals the mechanisms governing gingival immunity early in life, with the potential to affect oral health in adult life.

## Results

### Adaptive immunity establishes in the gingiva during weaning

To characterize the development of gingival immunity postnatally, the frequencies of various leukocytes in the developing gingiva were quantified. Gingival tissues were collected at weeks 1, 4, and 8 after birth, representing three critical developmental periods: neonatal, weaning, and adulthood, respectively. Upon separation of the epithelium and the lamina propria, the samples were subjected to flow cytometry examination. tSNE analysis display the expression of CD45 on the purified cells (Fig. 1A) and the total numbers of CD45+ leukocytes, indicating a significant increase of the leukocytes during weaning in both tissue parts. In addition, the frequencies of the various leukocyte subsets changed during postnatal development (Fig. 1B). Detailed analysis revealed that while the proportions of the innate leukocytes, neutrophils, and monocytes, decreased during weaning, the adaptive T and B lymphocytes increased in this period (Fig. 1B and Supplementary Fig. 1). CD4+ and CD8+ T cells were detected in the epithelium and lamina propria, while the majority of these lymphocytes in the epithelium express CD69 and CD103, in line with their characterization as tissue-resident cells (Fig. 1C)[12,13]. The expression of Ly6C, which indicate a state of activation of short-lived T cells[14], was upregulated in CD8+ T cells that were CD69neg CD103neg. With regards to CD4+ T cells, Ly6C expression was evident in the lamina propria but not in the epithelium. B cells were detected mainly in the lamina propria where they represent more than 40 to 55% of the leukocytes in weaned and adult mice, respectively (Fig. 1B). γδT cells were also identified in the epithelium and lamina propria, but their frequencies decreased during postnatal development in the epithelium (Fig. 1B). Of note, a population of innate lymphoid cells (ILCs) was observed in the gingiva, and while in the adult epithelium, the cells acquired a tissue-resident CD69+ CD103+ phenotype, in the adult lamina propria the majority were negative to these markers (Fig. 1D). Collectively, these findings demonstrate that gingival leukocytes largely acquire adult features during weaning, while further adaption takes place until adulthood.

### The microbiota has a limited effect on the establishment of gingival homeostasis

To examine the impact of the microbiota on gingival immunity, we compared the leukocytes in the gingiva of adult germ-free (GF) and specific pathogen-free (SPF) mice. The frequencies of total leukocytes and their major subsets were similar in GF and SPF mice (Fig. 2A, B). Nevertheless, further analysis revealed that the microbiota differentially regulates several leukocytes in the epithelium. Whereas the frequencies of neutrophils, γδT cells, and CD8+ T cells decreased, the percentages of the ILCs increased in the gingival epithelium (Fig. 2B). Next, it has been shown that during weaning the expression of IFN-γ and TNF-α is transiently upregulated in the intestine by the microbiota, a process termed a weaning reaction that is crucial to prevent susceptibility to inflammatory pathologies in the adult[15]. We thus quantified by RT-PCR analysis the expression of these cytokines in the gingiva of GF and SPF mice. Similar to the intestine, the expression of *Ifng* was upregulated during weaning, however, this was not mediated by the microbiota (Fig. 2C). Moreover, in GF mice the expression of *Ifng* was further elevated in the adult. Expression of *Tnfa* was also elevated in the adults in GF mice, yet the expression was already high in neonates. We next quantified the expression of *Il17a* and *Foxp3* because a balanced ratio of these molecules signifies mucosal homeostasis[16,17]. Both genes were upregulated during weaning in SPF and GF gingiva and the expression further increased in the adult GF mice. The *Il17a* /*Foxp3* ratio, however, was not significantly altered between the GF and SPF mice (Fig. 2C). These data suggest that the microbiota regulates immunological function mainly in the epithelium. Other immunological functions, however, appear to take place regardless of the microbiota, suggesting that other mechanisms are involved in the establishment of gingival immunity.

### The microbiota facilitates the differentiation of gingival LCs until weaning that have a potent migratory capacity to the lymph node

As the predominant APCs of stratified epithelia, LCs respond to various epithelial challenges by migrating to the draining lymph nodes (LN) where they can prime adaptive immunity[6]. To examine whether

gingival LCs play a role in the postnatal establishment of gingival immunity, we first characterized the differentiation kinetics of these cells in the developing gingiva. Gingival tissues were collected from mice at various weeks after birth and analyzed by flow cytometry. As depicted in Fig. 3A, CD11c⁺ MHCII⁺ cells (APC) were detected in the gingival epithelium during weaning, where a large part of them further expressed the LC-defining markers langerin and EpCAM. The APCs located in the lamina propria, on the other hand, failed to express the LC markers (Fig. 3A). The relative frequencies of LCs remain constant

in 6- and 8-week-old mice compared to 4-week-old mice, suggesting that LCs efficiently populate the gingiva until weaning. In line with this notion, immunofluorescence staining of gingival epithelial sheets collected from 4- and 8-week-old mice revealed a similar rich network of langerin and MHCII positive cells, which also displayed a typical dendritic morphology (Fig. 3B). To better understand the development of gingival LCs, we next examined the impact of the microbiota on the initial establishment of LCs during weaning. For this, gingival epithelial sheets were produced from 4-week-old GF and SPF mice for

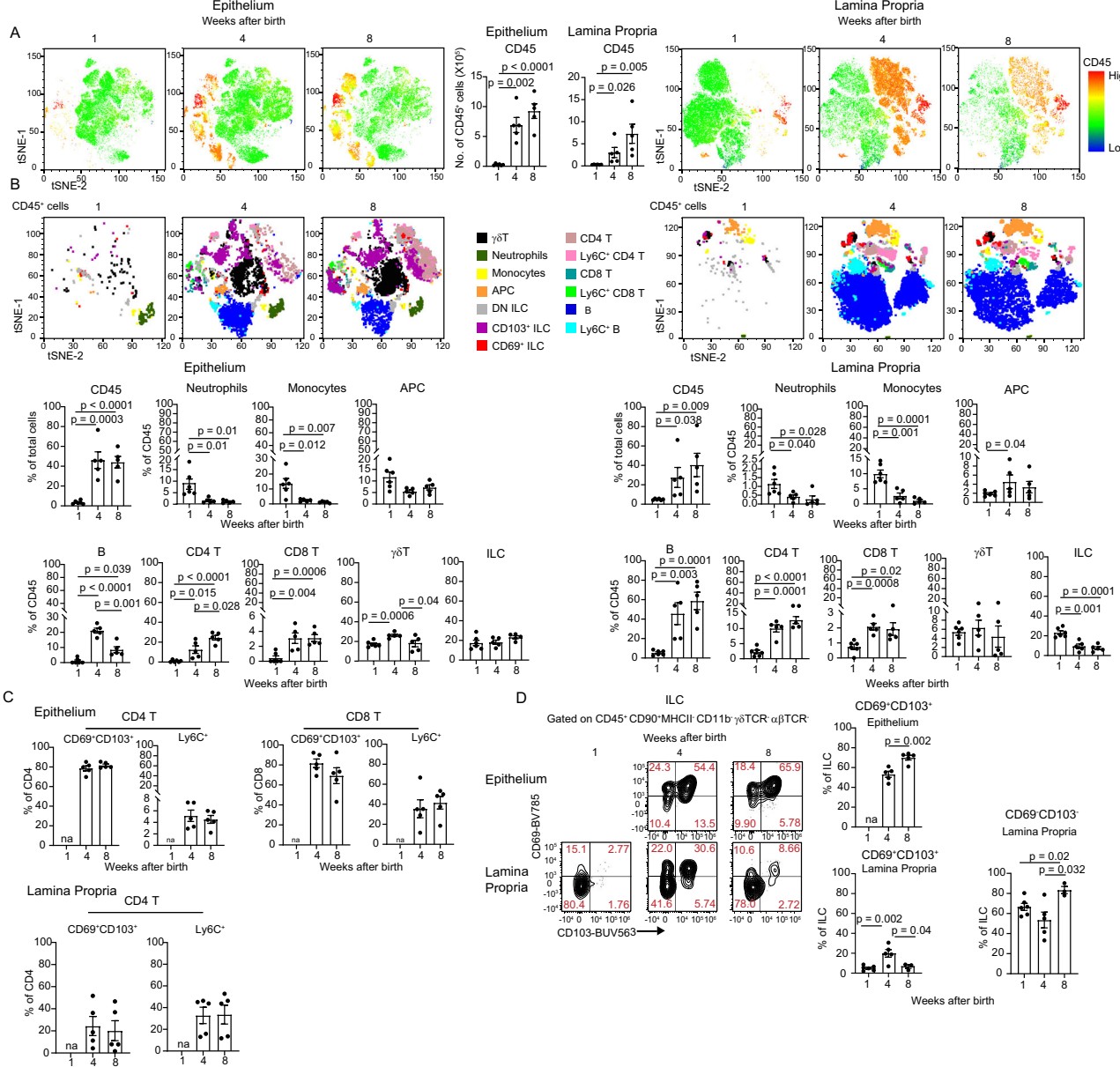

**Fig. 1 | Gingival adaptive immunity is established during weaning. A** tSNE flow cytometry plots display the expression of the leukocyte marker CD45 on total tissue cells and the total numbers of CD45⁺ leukocytes in the gingival epithelium and lamina propria at the indicated weeks after birth. Bar graph represents the mean leukocyte numbers + SEM (*n* = 5 mice). *p*-value from an ordinary one-way ANOVA test using GraphPad Prism. **B** tSNE plots show the main cells subsets of CD45⁺ leukocytes present in each tissue at the indicated weeks after birth. Representative data from two independent experiments. Graphs show the mean frequencies + SEM (*n* = 5 mice for weeks 4 and 8, *n* = 6 mice for week 1) of the main leukocytes of the gingiva at various weeks postnatally. Data from one out of two independent experiments are shown. *p*-value from an ordinary one-way ANOVA

test using GraphPad Prism. **C** Graphs show the mean frequencies + SEM (*n* = 5 mice) of CD69⁺ CD103⁺ or Ly6C⁺ CD4⁺ and CD8⁺ T cells in the epithelium and lamina propria at various weeks postnatally. Data from one out of two independent experiments are shown. *p*-value from an ordinary one-way ANOVA test using GraphPad Prism. **D** Representative flow cytometry plots and graphs show the mean frequencies + SEM of the ILCs expressing CD69 and CD103 in the epithelium (*n* = 5) and lamina propria (*n* = 6 mice for week 1, *n* = 5 mice for week 4, *n* = 3 mice for week 8) at various weeks postnatally. Data from one out of two independent experiments are shown. *p*-value from an ordinary one-way ANOVA test using GraphPad Prism. Source data are provided as a Source Data file.

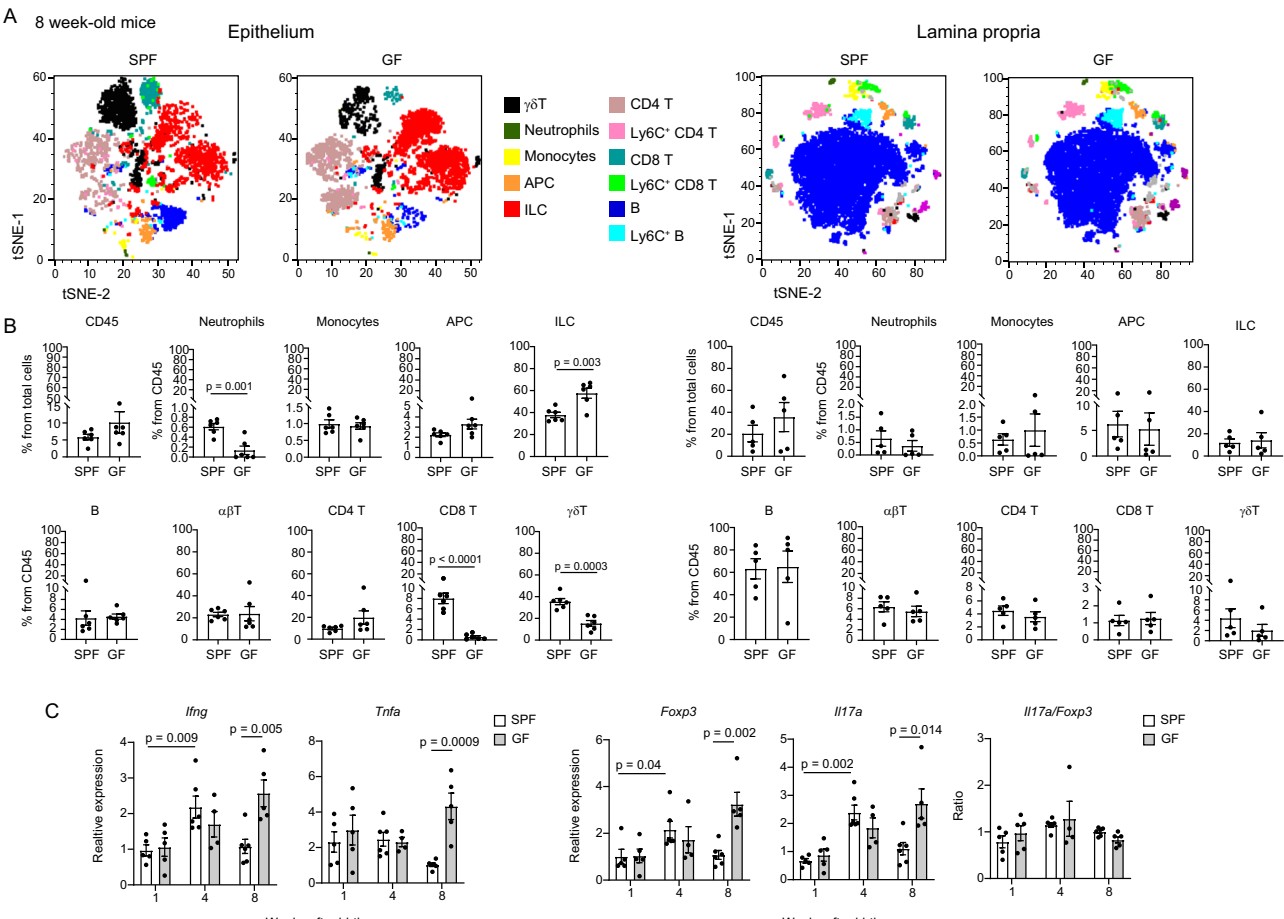

**Fig. 2 | The microbiota has a limited impact on gingival homeostasis. A** tSNE flow cytometry plots display the main subsets of leukocytes present in the gingival epithelium and the lamina propria of 8-week-old SPF and GF mice. Representative data from three independent experiments. *p*-value from two-tailed, unpaired *t*-test using GraphPad Prism. **B** Graphs show the mean frequencies + SEM (*n* = 5 for SPF, 6 for GF) of the various leukocytes in the gingival tissues of SPF and GF mice. Data from one out of three independent experiments are shown. *p*-value from a two-tailed, unpaired *t*-test using GraphPad Prism. **C** Relative expression of the noted immunological genes in the gingiva at various weeks after birth. Graphs present the transcript levels quantified by RT-PCR and normalized to 8-week-old mice depicted as the mean + SEM (*n* = 5 for SPF week 1 and GF weeks 1 and 8, *n* = 6 for SPF weeks 4 and 8, *n* = 4 for GF week 4). Representative data of two independent experiments. *p*-value from an ordinary one-way ANOVA test using GraphPad Prism. Source data are provided as a Source Data file.

immunofluorescence analysis. As depicted in Fig. 3C, significantly lower numbers of langerin-positive cells, representing LCs, were found in GF mice compared to SPF, particularly in epithelial areas close to the tooth. Furthermore, while the dendrites of SPF LCs were elongated and directed toward the putative region of the microbial plaque, the dendrites of GF LCs were short and without preferential directionality (Fig. 3D). The reduction in the frequencies of LCs in 4-week-old GF mice was also verified by flow cytometry (Fig. 3E). These results are in line with our previous observations demonstrating the importance of the microbiota to LC differentiation in adult mice[7]. Next, we asked whether the LCs residing in the gingiva at weaning can migrate to LNs under physiological conditions. For this, we painted the gingiva of 3-, 4-, and 8-week-old mice with FITC solution, and 3 days later the cervical LNs were collected and analyzed by flow cytometry (Fig. 3F). Gating on CD11c⁺MHCIIʰⁱ tissue-derived dendritic cells (DC), FITC-labeled DCs representing cells migrating from the gingiva, were mainly detected in the LNs of 4- and 8-week-old mice. Moreover, the relative frequencies of migratory LCs were significantly higher in 4-week-old mice compared to 8-week-old mice. It should be mentioned that other DCs also migrate from the gingiva, such as EpCAM⁺ langerin⁻ cells, which likely represent LCs that have lost langerin expression. The other EpCAM⁻ langerin⁻ cells might signify newly developing LCs or DCs located in the lamina propria, suggesting that multiple DC subsets respond to

immunological challenges occurring during weaning. These data indicate that the LC population is first established in the gingiva until weaning, with the microbiota regulating their development. Furthermore, gingival LCs migrate more frequently to the LNs during weaning, suggesting the existence of physiological conditions that induce their activation in this early period of life.

**Depletion of LCs has no impact on the initial colonization of the oral microbiota**

Since the microbiota regulates the differentiation of gingival LCs during weaning, we next asked whether the LCs affect the colonization of the oral microbiota. For this, we employed the langerin-DTR (diphtheria toxin receptor) mice enabling conditional ablation of langerin⁺ cells by administration of DT (diphtheria toxin)[18]. LCs were depleted from the third week of life until adulthood by weekly intraperitoneal administrations of DT (or PBS as a control). The oral microbiota was then sampled from individual adult mice for taxonomic analysis. As depicted in Fig. 4A. no differences were found in the oral microbial load of LC-depleted or non-depleted mice. Moreover, the diversity and composition of the microbiota were similar between these groups (Fig. 4B–D). This suggests that while the microbiota regulates the initial differentiation of LCs, LCs have no impact on the colonization of the microbiota.

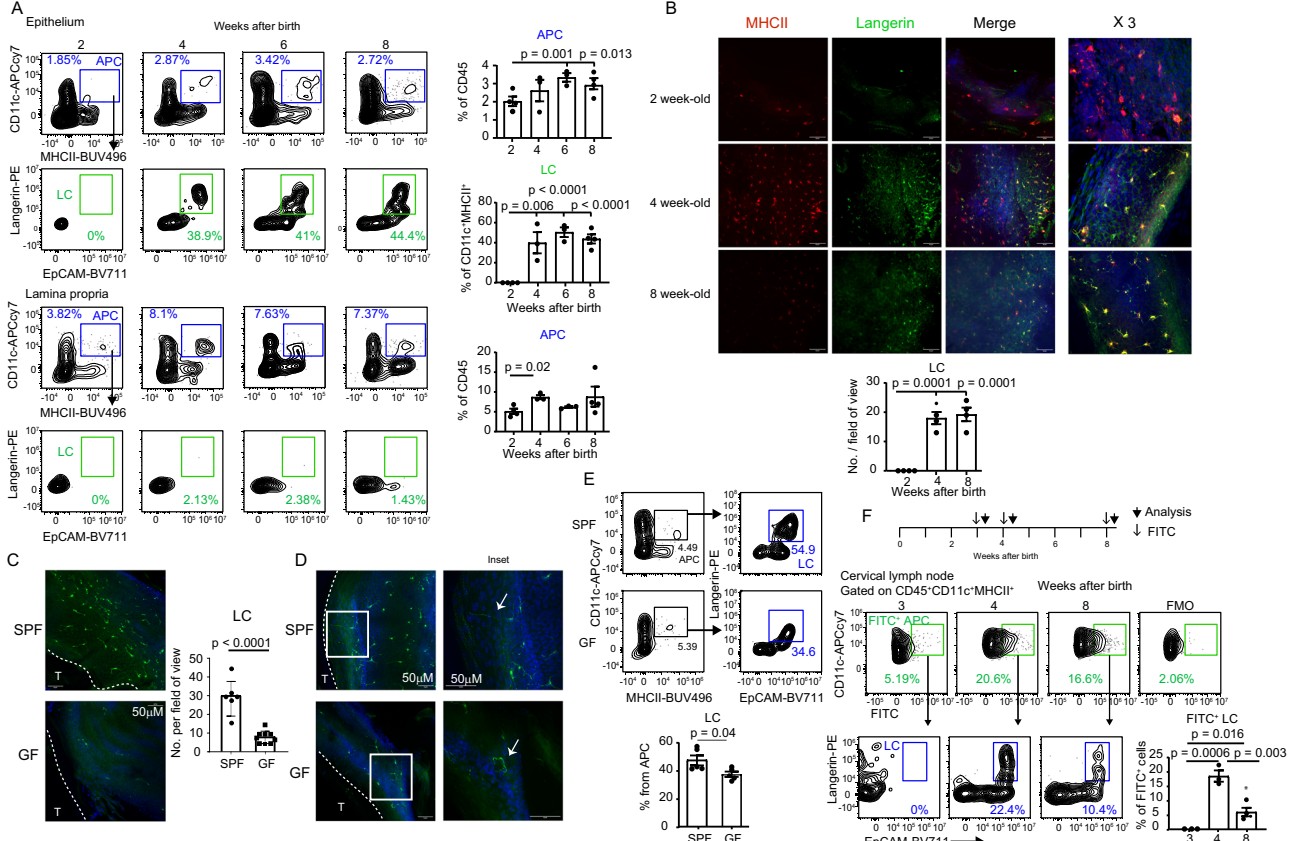

**Fig. 3 | LCs populate the gingival epithelium during weaning and have a potent migratory capability. A** Representative flow cytometry plots and graphs display the mean frequencies + SEM (*n* = 4 mice for weeks 1 and 8, *n* = 3 mice for weeks 4 and 6) of APCs and LCs in the gingival epithelium and lamina propria at various weeks after birth. Representative results from two independent experiments. *p*-value from an ordinary one-way ANOVA test using GraphPad Prism. **B** Wholemount immunofluorescence staining of gingival epithelial sheets prepared from 2, 4, and 8-weeks-old-mice with antibodies directed against MHCII (red), langerin (green), and with DAPI (Blue) for nuclear visualization. Representative images of two independent experiments. Scale bar, 50 μm. The graph presents the means of LC numbers per field of view + SEM (*n* = 4 mice). *p*-value from an ordinary one-way ANOVA test using GraphPad Prism. **C, D** Wholemount immunofluorescence staining of gingival epithelial sheets prepared from 4-week-old SPF and GF mice with

antibody directed against langerin (green) and DAPI (Blue). Representative images of three independent experiments. Scale bar, 50 μm. The graph presents the means of LC numbers per field of view + SEM (*n* = 6 for SPF, *n* = 9 for GF). *p*-value from a two-tailed, unpaired *t*-test using GraphPad Prism. **E** Representative flow cytometry plots and a graph display the mean frequencies + SEM (*n* = 5) of LCs in the gingival epithelium of 4-week-old SPF and GF mice. Representative results from two independent experiments. *p*-value from a two-tailed, unpaired t-test using GraphPad Prism. **F** Gingival tissues of B6 mice were painted with FITC solution 3, 4, and 8-weeks after birth, and 3 days later, the cervical LNs were collected. FACS plots and graphs show the frequencies + SEM (*n* = 3 for weeks 3 and 4, *n* = 4 for week 8) of FITC⁺ LCs among the APCs. FMO, fluorescence minus one. Representative results from two independent experiments. *p*-value from an ordinary one-way ANOVA test using GraphPad Prism. Source data are provided as a Source Data file.

## Early-life masticatory forces facilitate LC migration to the LNs

The limited impact of the microbiota on gingival homeostasis, and the lack of effect of LCs on oral microbial colonization, suggest that LCs react to another mechanism in this early period of life. We thus asked whether LCs respond to mechanical forces induced by initial mastication as they are capable of damaging the epithelium and LCs are strategically positioned to respond to such physiological challenges[19]. To address this, we provided the mice with a soft diet and examined how it impacts the LC population (Fig. 5A). As depicted in Fig. 5B, the soft diet specifically alters the frequencies of LCs in the gingival epithelium, as the percentages of gingival LCs but not total APCs were two times higher in mice fed with the soft diet compared to the solid diet. Accordingly, the total numbers of LCs in the gingiva were higher in mice fed with soft diet (Fig. 5C) and, respectively, the frequencies and numbers of migratory LCs in the cervical draining LNs were reduced (Fig. 5D–F). In addition, the expression of *Ccl20*, known to attract and retain LCs in the epithelium[7,20], was upregulated in mice provided with a soft diet (Fig. 5G). As this suggests that mechanical epithelial damage facilitates LC migration, we directly examined this by painting the gingiva with FITC solution and subsequently applying mechanical damage by brushing the gingiva with a cotton swab. Indeed, higher

levels of migratory FITC⁺ LCs were found in the cervical LNs of mice undergoing mechanical damage compared to FITC-painted-only mice (Supplementary Fig. 2). To examine if these migratory LCs can interact with T cells, we adoptively transferred to langerin-DTR mice CellTrace-labeled OT-I cells and then induced mechanical damage using a cotton swab soaked with the ovalbumin (OVA) antigen. Flow cytometry analysis of the cervical LNs revealed that the OT-I cells proliferated due to the damage, whereas depletion of the LCs reduced the intensity of the proliferation, suggesting that gingival LCs present antigen to T cells (Supplementary Fig. 3). Of note, the OT-I cells proliferate also in the absence of LCs, indicating that other DCs are involved in antigen presentation in this setting. Collectively, these findings suggest that early-life masticatory forces activate gingival LCs to migrate to the LNs.

## Early-life masticatory forces induce IL-17-independent alveolar bone-resorbing immunity

We next discover that the soft diet also alters gingival immunity, as the total gingival leukocytes were decreased, which was attributed to a reduction in the adaptive CD4⁺ T and B lymphocytes but not the innate cells (e.g., monocytes, neutrophils, and APCs)

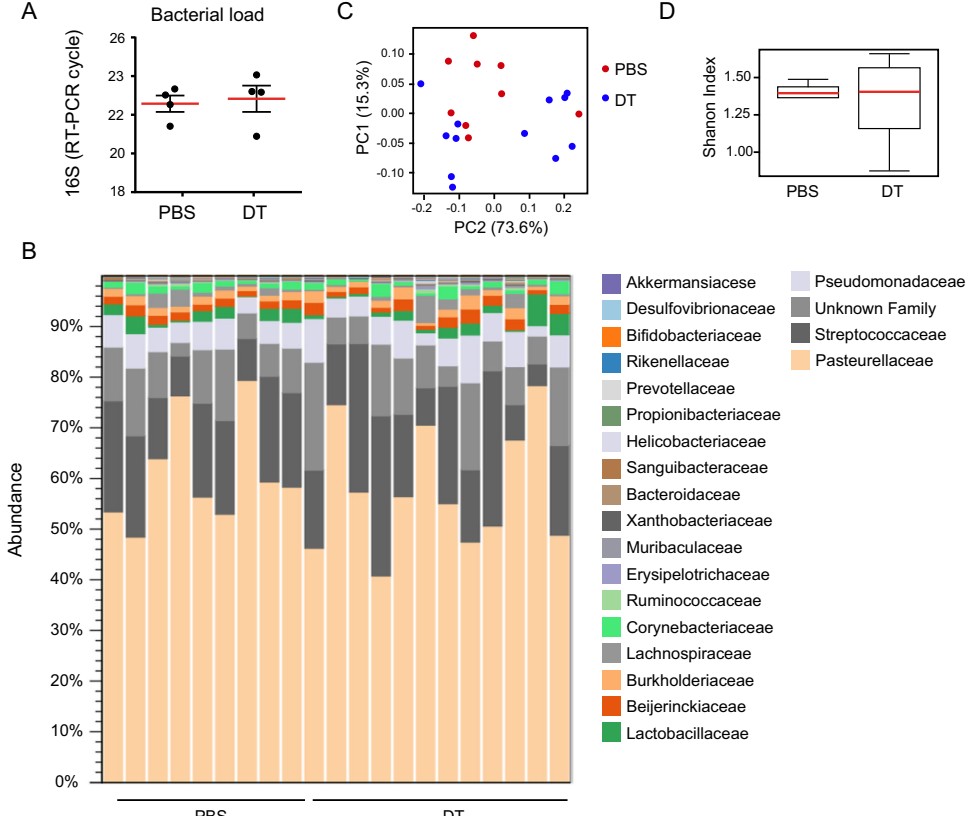

**Fig. 4 | Depletion of LCs during weaning has no impact on the oral microbiota.** Langerin-DTR mice were treated with DT or PBS to deplete LCs during weaning, and the oral microbiota was sampled from adult mice for taxonomic analysis. **A** Total oral bacterial load determined by quantitative RT-PCR of the mean expression of the 16 S rRNA gene + SEM ($n = 4$ mice). $p$-value from two-tailed, unpaired $t$-test using GraphPad Prism. **B** Histograms represent the distribution of sequences in operational taxonomic units (OTUs) assigned to each family. **C** Principal coordinates (PCs) analysis of weighted UniFrac distances based on 16S rRNA of Langerin-DTR mice treated with DT or PBS. **D** Alpha diversity plot representing taxa richness in samples of both groups of mice. Data presented as standard boxplot, the median denoted as a central horizontal line in the box and the whiskers covering the data within ±1.5 IQR ($n = 9$ for PBS, $n = 12$ for DT). Taxonomic data was pooled from two independent experiments. Source data are provided as a Source Data file.

(Fig. 6A, B). We next quantified by RT-PCR the expression of genes related to the weaning reaction and tissue homeostasis. As depicted in Fig. 6C, reduced expression of *Ifng* and *Tnfa* was observed in 4-week-old mice fed with a soft diet, yet the expression of *Tnfa* but not *Ifng* returned to normal levels in the adult. Expression of *Foxp3* and *Il17a* resembles those of *Ifng* and *Tnfa*, accordingly (Fig. 6C), suggesting that masticatory forces rather than the microbiota (Fig. 2) induce the expression of these key immunological genes. We then examined if the soft diet impacts the alveolar bone of 8-week-old mice. Quantification of the volume of the residual alveolar bone revealed elevated bone volume in mice fed with the soft diet compared to the regular solid diet (Fig. 6D). The bone trabecular properties were also altered due to the soft diet, demonstrating the considerable impact of early-life mastication on the alveolar bone. Accordingly, higher numbers of TRAP+ cells were detected in gingival cross-sections of mice fed with a solid than soft diet (Fig. 6E). To test if the early-life bone loss induced by the hard diet is mediated by IL-17 as reported in adult life[4], we provided a soft diet to *Il-17a*-/- mice. A micro-computed tomography (μCT) analysis indicated that similar to wild-type mice, the soft diet resulted in an increased volume of the alveolar bone despite the lack of IL-17 (Fig. 6F). Further analysis revealed that this was due to elevated trabecular thickness rather than the number of trabeculae. These data suggest that mastication-induced immunity associates with early-life bone loss, which, in contrast to adult life, is not mediated by IL-17.

## Depletion of LCs during weaning dysregulates gingival transcriptomic signature associated with epithelial integrity and homeostatic immunity

We next investigated the role of LCs on gingival homeostasis during early life. LCs were depleted from langerin-DTR mice during the weaning period by administrating DT or PBS on days 17, 24, and 31 after birth, and the repopulation of LCs was analyzed using flow cytometry (Fig. 7A). LCs were completely absent in 4-week-old mice, which then gradually repopulated the gingiva, restoring their normal frequencies in the eighth week of life (Fig. 7B). To explore the impact of LC ablation on the gingiva in an unbiased manner, we profiled the global gene expression of gingival epithelial cells by RNA sequencing (RNAseq). The hierarchical clustering and principal component analysis (PCA) indicated a significant difference between the epithelial cells purified from LC-depleted and non-depleted mice (Fig. 7C, D). As depicted in Fig. 7E, the analysis of gene set enrichment analysis (GSEA) revealed that the absence of LCs upregulated pathways associated with tissue/epithelial repair and epithelial integrity such as oxidative phosphorylation, MYC, p53, and DNA repair[21-24]. TNFα signaling via NFκB was also upregulated, a pathway known to mediate epithelial cell survival[25]. An upregulation of Wnt/β-catenin signaling was detected, which was shown to play a role in epithelial integrity[26] and gingival regeneration[27]. Similarly, hypoxia-related signaling was increased, a pathway controlling healthy barrier function and wound healing[28,29] as well as gingival function with age[30]. Various pathways were also

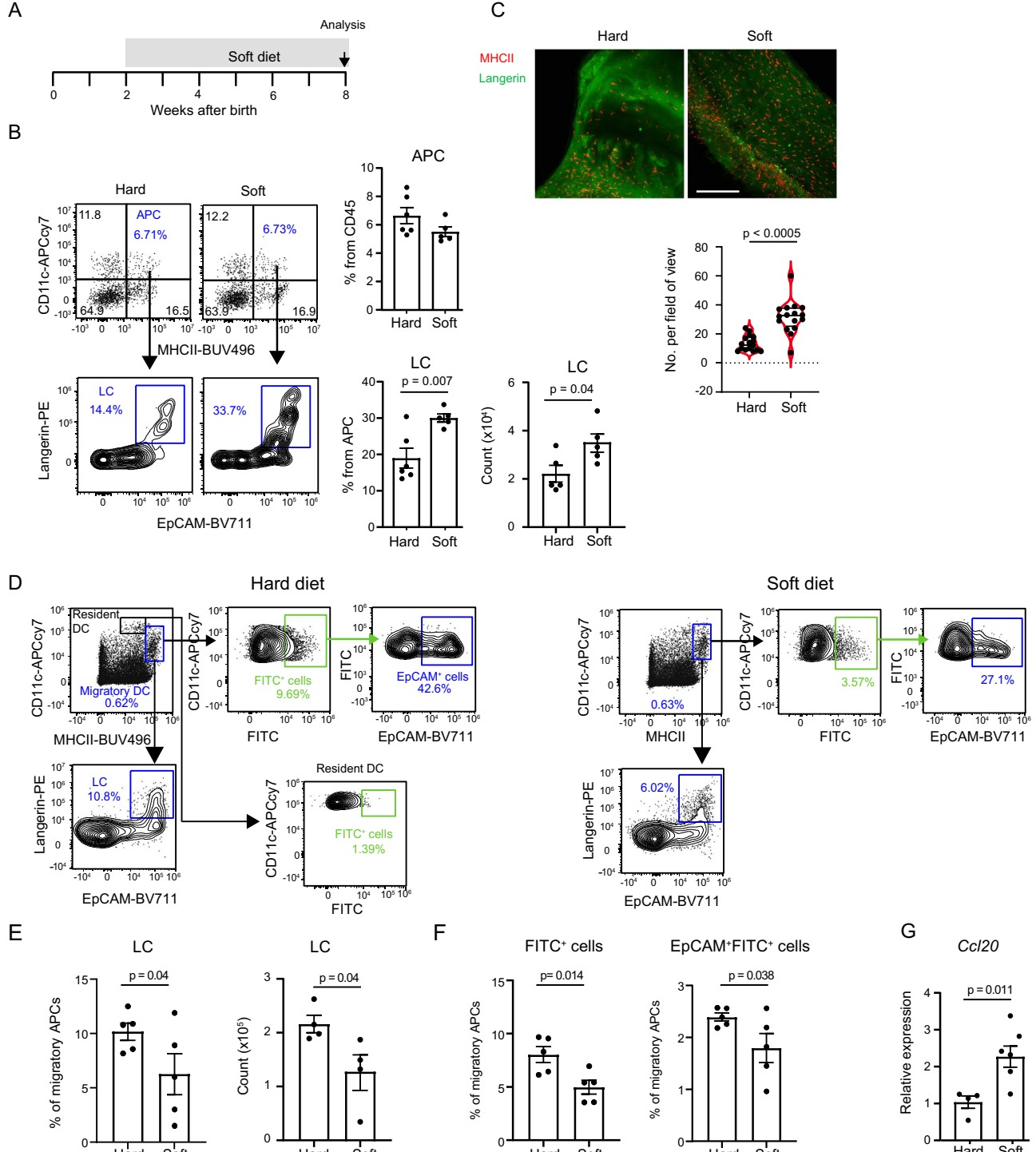

**Fig. 5 | Early-life masticatory forces facilitate LC migration to the LNs. A** Scheme of the experimental set-up to diminish masticatory forces by giving mice a soft diet from the second to the eighth week of life. **B** Mice were fed with a soft or solid diet from the second week of life, and at six weeks old, the gingiva was analyzed by flow cytometry. Representative FACS plots and graphs present the mean frequencies + SEM ($n = 5$ hard diet, $n = 6$ soft diet) and total numbers of gingival APCs and LCs. Representative data from two independent experiments. *p*-value from a two-tailed, unpaired t-test using GraphPad Prism. **C** Representative immunofluorescence images of gingival epithelial sheet prepared from mice fed with hard or soft diet and stained against MHCII and langerin to visualize LCs. Data presented the number of LCs as standard violin plot, the median denoted as a central horizontal line in the box and the whiskers covering the data within ±1.5 IQR ($n = 16$ fields from 3 mice). *p*-value from a two-tailed, unpaired t-test using GraphPad Prism. **D–F** Mice were fed with a soft or solid diet from the second week of life, and at six

weeks old, the gingiva was painted with FITC solution and two days later the cervical LNs were analyzed by flow cytometry. Representative FACS plots (**D**) and graphs (**E**) present the mean frequencies + SEM ($n = 5$ mice) and total numbers ($n = 5$ mice) of LCs among migratory APCs. **F** Graphs show the mean frequencies of FITC⁺ cells and EpCAM⁺ FITC⁺ cells (representing migratory LCs) among migratory APCs + SEM ($n = 5$ mice). Representative data from two independent experiments. *p*-value from a two-tailed, unpaired t-test using GraphPad Prism. **G** Relative expression of the *Ccl20* in the gingiva of mice fed with a soft or solid diet. Graphs present the transcript levels quantified by RT-PCR and normalized to mice fed with a solid diet depicted as the mean + SEM ($n = 4$ hard diet, $n = 6$ soft diet). Representative data from two independent experiments. *p*-value from a two-tailed, unpaired t-test using GraphPad Prism. Source data are provided as a Source Data file.

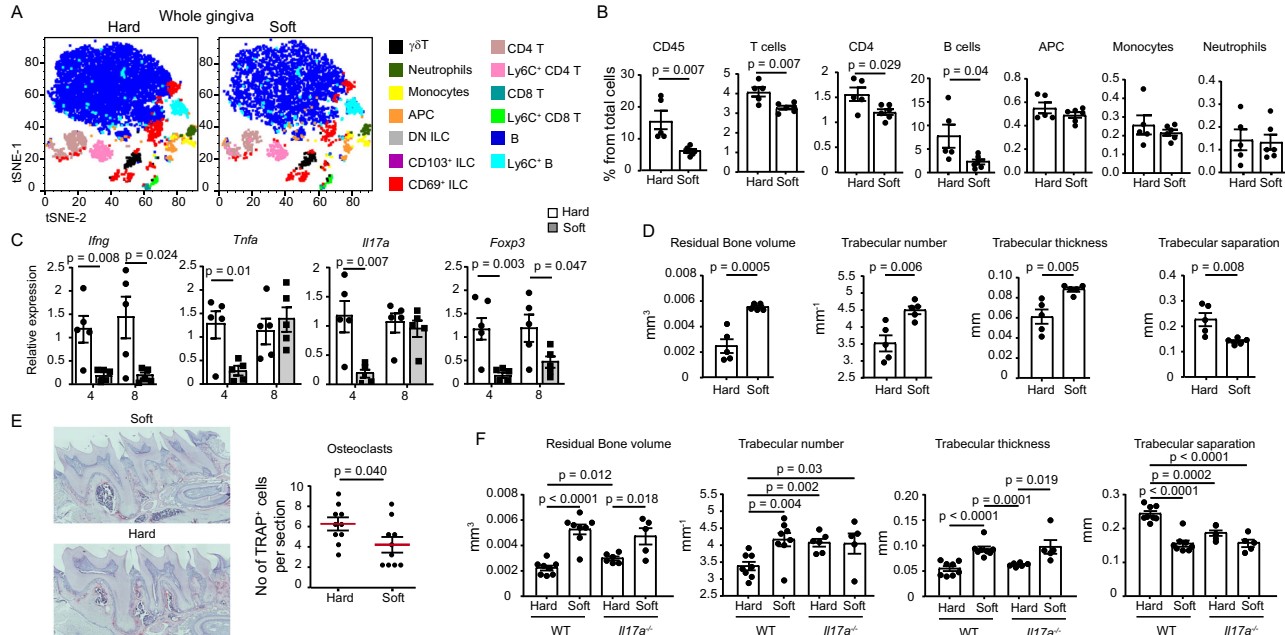

**Fig. 6 | Early-life masticatory forces alter gingival immunity and induce IL-17-independent alveolar bone loss. A** tSNE flow cytometry plots display the main subsets of leukocytes present in the gingiva (epithelium and lamina propria) of adult B6 mice receiving a soft or solid diet from the second week of life. *p*-value from a two-tailed, unpaired *t*-test using GraphPad Prism. **B** Graphs show the mean frequencies + SEM (*n* = 5 hard diet, *n* = 6 soft diet) of the various leukocytes in the gingiva. Data from one out of two independent experiments are shown. *p*-value from a two-tailed, unpaired *t*-test using GraphPad Prism. **C** Relative expression of the note genes in the gingiva of mice fed with a soft or solid diet. Graphs present the transcript levels quantified by RT-PCR and normalized to mice fed with a solid diet depicted as the mean + SEM (*n* = 5). Representative data from two independent experiments. *p*-value from a two-tailed, unpaired *t*-test using GraphPad Prism. **D** μCT analysis of the maxilla of adult B6 mice receiving a soft or solid diet prior to

weaning. The graphs show the mean alveolar bone volumes + SEM (*n* = 6) as well as the noted trabecular parameters. Representative data from two independent experiments. *p*-value from a two-tailed, unpaired t-test using GraphPad Prism. **E** Representative gingival cross-sections from adult mice fed with a soft or solid diet. The graph shows the mean numbers + SEM of TRAP-positive osteoclasts per cross-section (*n* = 10 sections from three mice). Representative data from two independent experiments. *p*-value from a two-tailed, unpaired *t*-test using Graph-Pad Prism. **F** *Il17a⁻/⁻* mice were treated with soft or solid diets as described above. The graphs show the mean alveolar bone volumes + SEM (*n* = 8 for WT, *n* = 5 for *Il17a⁻/⁻*) and the trabecular parameters using μCT analysis. Representative data from three independent experiments. Scale bar, 50 μm. *p*-value from an ordinary one-way ANOVA test using GraphPad Prism. Source data are provided as a Source Data file.

downregulated by the LC depletion such as elongation factor 2 (E2F) which plays a critical role in epithelial repair via facilitating local inflammatory responses and rates of re-epithelialization[31]. The decrease of the mitotic spindle and G2/M signaling further suggest an impact of LC on epithelial homeostasis as these pathways control polarity and intercellular forces of epithelial cells, respectively[32,33]. Hedgehog signaling, which involves the maintenance of homeostasis in the mucosal epithelium[34], was also downregulated. To verify in vivo the impairment of epithelial integrity due to LC ablation, we examined epithelial permeability by applying FITC solution on the gingiva of anesthetized mice and then quantifying its penetration to the epithelium. As depicted in Fig. 7F, whereas FITC was restricted to the keratin layer in PBS-treated mice, ablation of LCs resulted in substantial penetration of the dye to the epithelium that even labeled deeper gingival tissues. Of note, the increased permeability was not due to dysregulated expression of Claudin 1 (*Cld1*), Claudin 4 (*Cld4*), and tight junction proteins 1 (*Tjp1*, also known as ZO-1) (Supplementary Fig. 4), suggesting that another mechanism rather than the integrity of tight junction-related molecules might be involved. Next, the GSEA analysis further revealed that LC ablation reduced several immunological pathways such as IFN-α and IFN-γ signaling that was reported to control mucosal epithelial homeostasis including in the gingiva[16,35]. A reduction in IL-2/STAT5 signaling was also detected, an important immunological pathway promoting Treg cell differentiation while limiting Th17 cells[36,37]. Collectively, these findings suggest that ablation of LCs impairs gingival epithelial integrity and reduces

homeostatic immunity, thus facilitating the expression of tissue repair mechanisms.

## LCs control the establishment of gingival immunity and mediate early-life alveolar bone loss

Thus far, masticatory forces found to induce the migration of LCs to the LNs, and the RNAseq analysis further implies that LCs support the development of immunity to maintain epithelial integrity. Therefore, we next address the direct impact of LCs on gingival immunity by depleting these cells during early life. As depicted in Fig. 8A, tSNE analysis indicated no change in the types of leukocytes present in the gingival epithelium of LC-depleted or non-depleted mice (Fig. 8B). Nevertheless, further analysis revealed a significant decrease in the frequencies of CD4⁺FOXP3⁺ Treg cells in LC-depleted mice (Fig. 8B). Similar results were obtained when the depletion of LCs was maintained until adulthood (Fig. 8C). We then quantified the production of IL-17A and IFN-γ by gingival CD4⁺ T cells upon ex vivo stimulation. In agreement with the RNAseq analysis, we detected reduced production of IFN-γ by CD4⁺ T cells due to the absence of LCs while IL-17A-producing CD4⁺ T cells were increased (Fig. 8D). Analysis of the innate immune cells, neutrophils, and monocytes, revealed that both populations were reduced in the gingiva of LC-depleted mice (Fig. 8E). Accordingly, *Cxcl1* and *Ccl2* expression, chemoattractants of both cell types, respectively, were decreased due to the depletion, whereas *Cxcl2*, which also attracts neutrophils was not affected (Fig. 8F)[38,39]. Since early-life mastication-induced immunity associates with alveolar bone loss, we asked whether depletion of LCs influences the alveolar

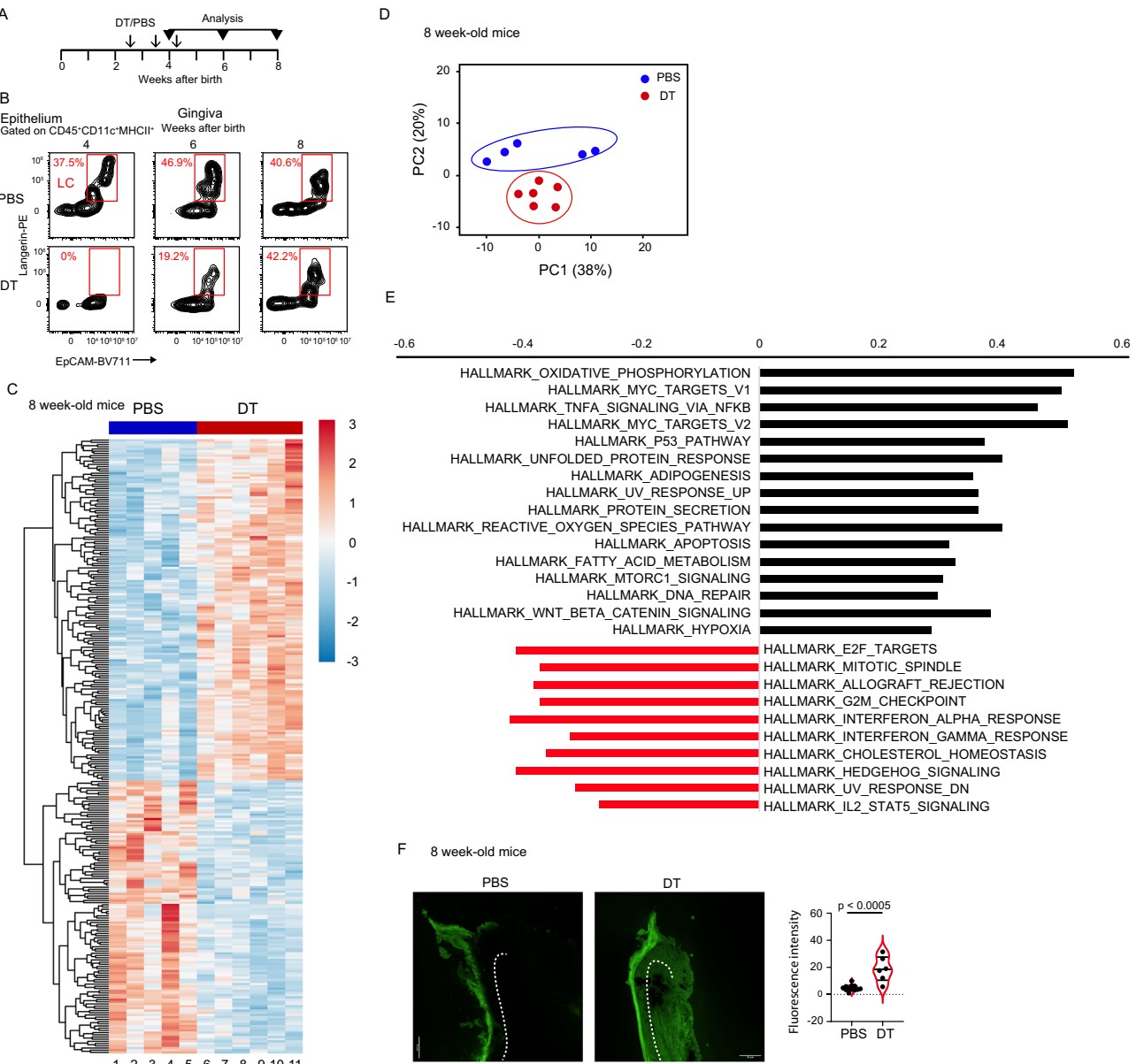

**Fig. 7 | Depletion of LCs during weaning dysregulates the cellular and immunological function of the adult gingiva. A** Experimental setting employed to deplete langerin-positive cells from Langerin-DTR mice using DT or PBS as a control. **B** Representative flow cytometry plots demonstrating the repopulation of gingival LCs upon the noted depletion strategy. **C**–**E** Epithelial cells were collected from adult Langerin-DTR mice after administration of DT (*n* = 6) or PBS (*n* = 5) as described in **A**, and subjected to global gene expression analysis. **C** Hierarchical clustering of the genes differentially expressed in the naive and the LC-depleted epithelial cells. **D** PCA of the most variable transcripts expressed by the epithelial cells from the different groups. **E** Significantly upregulated and downregulated gene pathways identifies by GSEA analysis among the various groups of epithelial cells (familywise error rate [FWER] <0.05). **F** FITC solution was applied to the gingiva of 8-weeks-old langerin-DTR mice treated with DT or PBS as described in **A**. Representative gingival cross-sections show the localization of FITC in the tissue upon topical application of FITC solution. Data presented the quantification of fluorescence intensity as standard violin plot, the median denoted as a central horizontal line in the box and the whiskers covering the data within ±1.5 IQR (*n* = 6 for DT, *n* = 12 for PSB). Scale bar, 50 μm. *p*-value from a two-tailed, unpaired *t*-test using GraphPad Prism. Source data are provided as a Source Data file.

bone in adulthood. Indeed, μCT analysis demonstrated a reduced exposed root area in 8-week-old LC-depleted mice (Fig. 8G). A 3D analysis further showed an increased alveolar bone volume in this group (Fig. 8H). Of note, quantification of the bone trabecula indicated that depletion of LCs did not change the basic structure of the alveolar bone (i.e. trabecular numbers, thickness, and separation). In line with the reduced alveolar bone loss observed in LC-depleted mice, and the decrease of monocytes that give rise to osteoclasts[40], lower numbers of osteoclasts were found in gingival cross-sections prepared from these mice (Fig. 8I). Collectively, our data suggest that epithelial damage caused by early-life masticatory forces activate LCs to induce

gingival immunity required to protect the epithelium but is also associated with alveolar bone loss.

## Discussion

This study demonstrates the essential role of LCs in controlling gingival immunity early in life. LCs efficiently populate the gingival epithelium before weaning, enabling the sealing of the epithelium, and inducing local immunity in response to epithelial damage caused by the masticatory forces exerted for the first time in life. Such immune responses are associated with alveolar bone resorption, which is considered a natural bone loss. In contrast to adult life, in which

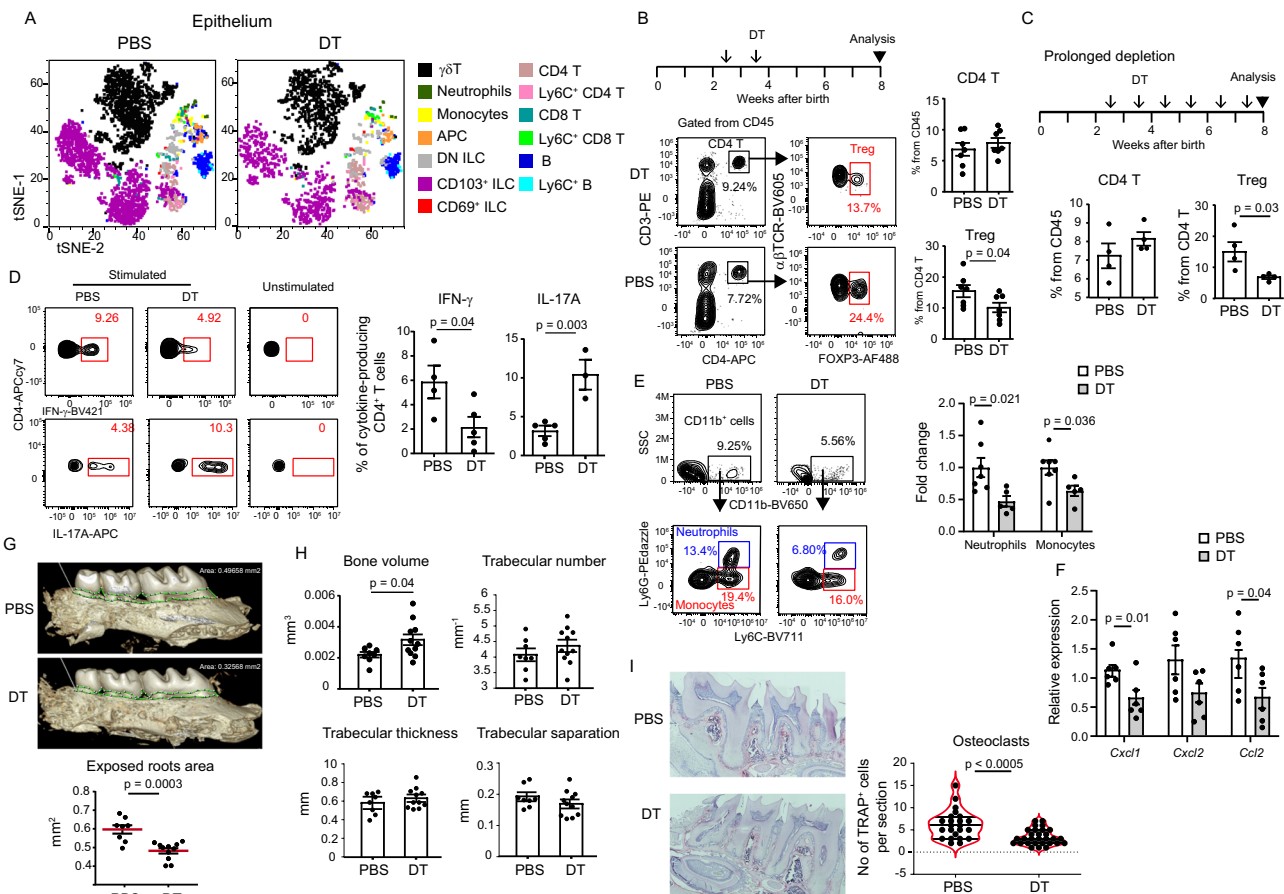

**Fig. 8 | LCs control gingival immunity after birth associated with alveolar bone loss. A** tSNE flow cytometry plots display the main subsets of leukocytes present in the gingival epithelium of Langerin-DTR mice treated with DT or PBS to deplete gingival LCs during weaning. **B, C** Gingival LCs were depleted during weaning (**B**) or until adulthood (**C**) as indicated in the schematics of the experimental settings. Representative flow cytometry plots and graphs show the mean frequencies + SEM ($n = 7$ for weaning, $n = 4$ for adulthood) of the Treg cells in the adult gingiva. Data were pooled from two independent experiments. *p*-value from a two-tailed, unpaired *t*-test using GraphPad Prism. **D** Representative flow cytometry plots and graphs display the mean frequencies + SEM of IFN-γ ($n = 4$ for PBS, $n = 5$ for DT) or IL-17A ($n = 5$ for PBS, $n = 3$ for DT) positive gingival CD4[+] T cells of adult mice depleted of LCs during weaning upon stimulation with PMA. Representative data from two independent experiments. *p*-value from a two-tailed, unpaired *t*-test using GraphPad Prism. **E** Representative flow cytometry plots and graphs display the mean frequencies + SEM ($n = 7$ for PBS, $n = 5$ for DT) of gingival monocytes, and neutrophils in adult mice depleted of LCs during weaning. Representative data from two independent experiments. *p*-value from a two-tailed, unpaired *t*-test using

GraphPad Prism. **F** Relative expression of the note genes in the gingiva of LC-depleted mice. Graphs present the transcript levels quantified by RT-PCR and normalized to PBS-treated mice depicted as the mean + SEM ($n = 6$). Data were pooled from two independent experiments. *p*-value from a two-tailed, unpaired *t*-test using GraphPad Prism. **G, H** μCT images and graphs of the adult maxilla of mice depleted of LC during weaning show the (**G**) exposed root area, the (**H**) alveolar bone volume, and trabecular parameters ($n = 8$ for PBS, $n = 11$ for DT). Data were pooled from two independent experiments. *p*-value from a two-tailed, unpaired *t*-test using GraphPad Prism. **I** Representative gingival cross-sections from adult mice show TRAP-positive cells in DT or PBS-treated mice during weaning. Data presented the numbers of TRAP-positive osteoclasts per cross-section as standard violin plot, the median denoted as a central horizontal line in the box and the whiskers covering the data within ±1.5 IQR The graph shows the ($n = 23$ sections for PBS, $n = 29$ sections for DT, collected from 3 mice). Representative data from two independent experiments. Scale bar, 50 μm. *p*-value from a two-tailed, unpaired *t*-test using GraphPad Prism. Source data are provided as a Source Data file.

mastication-induced bone loss is mediated by IL-17-producing CD4[+] T (Th17) cells[4], IL-17 is not involved in this process postnatally. Instead, IFN-γ-producing T cells (Th1) induced by the LCs may mediate early-life bone loss, as Th1 cells were reported to facilitate bone loss in a setting of periodontal inflammation[41,42]. While the flow cytometry analysis reveals that LCs inhibit the generation of Th17 cells and promote Th1 responses, the RNAseq analysis identifies changes only in IFN-γ but not IL-17-related signaling due to the depletion of LCs. This could be explained by the notion that IL-17 is mainly produced by γδT cells in the young murine gingiva and involved similar signaling as the CD4[+] T cells[43], thus the secretion by γδT cells might obscure the effect of LCs on the CD4[+] T cells. Regardless, whether LCs are also involved in mastication-induced bone loss in adult life is unclear. The dependence of the mastication-induced Th17 cells on interaction via the T-cell receptor[4], and the capability of mechanical damage to facilitate the

migration of LCs to the LNs, suggest that this could be the case. Moreover, prolonged depletion of LCs for 16 weeks reduces Treg cells and elevates IL-17 expression in the adult gingiva which is accompanied by bone loss[7]. Yet, while the present work also detected reduced Treg cells and elevated Th17 cells when LCs were depleted in early life, this did not translate to alveolar bone loss. Therefore, this proposes that mastication-induced bone loss involves distinct mechanisms in early and adult life, the former is IL-17-independent while the latter depends on IL-17.

The reported dispensability of the microbiota to the development of gingival Th17 cells represents a specific mechanism of the gingival epithelial barrier[4]. Nevertheless, the present study reveals that the microbiota does regulate gingival epithelial immunity in early life. The microbiota also facilitates the sealing of the oral epithelium after birth and upregulates the expression of its immunological pathways[2,16].

While these findings demonstrate the critical impact that the epithelium has on gingival immunity, it proposes a 'segregation of labor' between the microbial and mechanical challenges throughout life. Accordingly, the microbiota modulates epithelial gingival immunity until weaning, while masticatory forces begin at this time and become more influential on gingival immunity with age. This may explain the higher expression of various cytokines in the adult gingiva of GF mice compared to SPF, as in the absence of the microbiota, the epithelium is not optimally sealed during weaning, resulting in increased damage to the epithelium by mastication that eventually develops elevated immune responses. Additionally, interactions between commensal microbiota and the multiple cell types involved in wound healing regulate the immune response and promote barrier restoration[44,45]. Thus, the lack of the microbiota might delay the healing of epithelial damage induced by mastication, resulting in higher activation of the gingival immune system.

Unlike neutrophils and γδT cells that are reduced in the GF epithelium, in agreement with previous studies[2,17,46], the frequencies of ILCs increase. This suggests that ILCs respond to epithelial damage induced by mastication, similar to their reported role in tissue repair in other mucosae[47,48]. Of note, the microbiota inversely regulates the frequencies of the epithelium-resident CD8[+] T cells and ILCs and in fact, the formers are virtually absent in the GF epithelium. The murine oral epithelium contains a large population of ILCs, while the majority are ILC1 with cytotoxic capabilities, including the elimination of virally infected cells[49–52]. It is thus possible that ILCs respond to mastication-induced damage and epithelial infection; however, the CD8[+] T cells take over the latter activity upon exposure to the microbiota.

The microbiota is also capable of controlling the differentiation of gingival LCs during weaning. This could be attributed to the ability of the microbiota to regulate the expression of TGF-β1 and BMP-7, cytokines instructing mucosal LC development, which their expression is upregulated before weaning[7]. Interestingly, TGF-β1 expression in intestinal epithelial cells is activated by butyrate produced by gut commensal bacteria[53]. In the oral cavity, there are many butyrogenic bacteria that could regulate TGF-β1 levels by oral epithelial cells and consequently LC differentiation[54]. Moreover, the microbiota was reported to regulate the level of GM-SCF, a cytokine driving DC and LC differentiation, while macrophage-mediated sensing of microbial signals controls GM-CSF levels[55].

Our data demonstrate that gingival LCs sense and respond to epithelial damage by migrating to the LNs to prime T cells. Oral and skin epithelial cells that are stressed rapidly upregulate ligands for the lymphocyte activation receptor natural killer group 2D (NKG2D), resulting in the migration of LCs to the LNs, followed by the appearance of αβT cells that eliminate stressed epithelial cells[56–58]. A similar response also takes place after a minor injury, such as repetitive tape-stripping of the epidermis[59], which likely resembles the minor damage induced by ongoing masticatory forces. Under such mild injury conditions, epidermal LCs were shown to penetrate the tight junctions that link epithelial cells, enabling surveilling of the epithelium[60]. While the induction of Th1 cells by LCs facilitates inflammation that either enhances or delays tissue repair, the Treg cells are beneficial as they play a protective role in barrier repair after injury[61]. In this regard, the expansion of the Th17 cells over the Th1 cells with age is beneficial, as Th17 cells have superior tissue protective capabilities[62].

In humans, weaning typically begins around six months of age, a period when LCs are present in the oral epithelium[63], suggesting that primary mastication and LCs may also shape human periodontal immunity early in life. However, human deciduous teeth are eventually replaced by permanent teeth and the effect of this process on gingival immunity is unclear. Regardless, we demonstrate that LCs inhibit the development of Th17 during early life, a subset of T cells that have been reported to progressively expand with age in the gingiva of mice and humans due to ongoing masticatory forces and mediate alveolar bone loss[4]. This suggests that LCs may set the bar for adult life in humans because the lower the initial percentage of Th17, the slower bone loss will be later in life.

In summary, this study demonstrates that microbial colonization and masticatory forces shape the initial establishment of adaptive immunity in the gingiva. It further highlights the critical impact of the epithelium and its resident LCs in this process. Besides shedding light on the mechanisms orchestrating gingival immunity early in life, this work also increases our understanding of how early-life intervention (e.g., antibiotics, soft diet) might modulate oral immunity that might have clinical implications in adult life.

## Methods
### Mice
C57BL/6 (B6), langerin-DTR, *B6.129P2-Il17*[atm1Yiw] *(Il17a*[−/−]*)* male mice were bred and maintained in the central animal facility at the Hebrew University Faculty of Medicine (Jerusalem, Israel). The mice were maintained under SPF conditions and analyzed at various ages as described in the text. All animal protocols were approved by the Hebrew University Institutional Animal Care and Use Committee (IACUC). Germ-free (GF) B6 mice were maintained in sterile isolators at the Weizmann Institute of Science, and the GF studies were approved by the IACUC of the Weizmann Institute of Science

### Antibodies
The following fluorochrome-conjugated monoclonal antibodies and the corresponding isotype controls were purchased from BioLegend (San Diego, CA, USA): γδTCR (GL3), αβTCR(H57-597), I-A/I-E (M5/114.15.2), CD45.2 (104), langerin (4C7), Ly6G (1A8), Ly6C (HK1.4), CD3 (17A2), B220 (RA3-6B2), CD4 (GK1.5), CD11b (M1/70), CD11c (N418), FOXP3 (MF-14), NK1.1 (PK136), and CD69 (H1.2F3). For extracellular staining a 1:250 dilution was used and for intracellular staining 1:100. The following antibodies were used for immunofluorescence staining: Purified rabbit anti-mouse CD207/Langerin polyclonal (aa26-44) antibody (LSBio) and purified rat anti-mouse I-A/I-E (M5/114.15.2) (BioLegend) (1:100 dilution for both antibodies).

### Isolation of tissue leukocytes
The gingival tissues and lymph nodes were excised. In some of the experiments, gingival tissues were incubated in 1 mL of 4 mg/ml Dispase in PBS + 2% FCS until fully distended for 30 min. The epithelium and sub-epithelium were carefully separated using forceps and a binocular microscope. Tissues were then minced and treated with a Collagenase type II (2 mg/mL; Worthington Biochemicals) and DNase I (1 mg/mL; Sigma) solution in PBS plus 2% FCS for 25 min at 37 °C in a shaker bath. A total of 20 μL of 0.5 M EDTA per 2 mL sample was added to the digested tissues and incubated for an additional 10 min. The cells were then washed, filtered with 70-μM filter, stained with antibodies, run in Aurora (Cytek) flow cytometer, and further analyzed generating plots and tSNE plots using FlowJo software (BD Biosciences).

### Immunofluorescence staining
For wholemount staining gingival tissues were incubated in 1 mL of 4 mg/ml Dispase in PBS + 2% FCS until fully distended for 30 min and epithelium was carefully separated using forceps and binocular microscope. Tissues were then fixed in ice-cold 95% ethanol for 40 min. For frozen section staining, the mandibles were fixed overnight at 4 °C in 4% paraformaldehyde/PBS solution, decalcified for 2–3 weeks in EDTA, embedded in OCT, and cryo-sectioned into 10-μm-thick sections. The cross sections, as well as the whole tissues, were washed 3 times in PBS, blocked in blocking buffer (5% FCS, 0.1% Triton X-100 in PBS) for 1 h at room temperature, and incubated with a primary antibody overnight at 4 °C. Following 3 washing steps in PBS, the samples were incubated with a secondary antibody diluted 1:200 in

blocking buffer for 2 h at room temperature, washed 3 times, stained with DAPI and mounted. For paraffin sections, the salivary glands were fixed overnight at 4 °C in 4% paraformaldehyde/PBS solution, and then the tissues were dehydrated using 70%, 80%, 90%, and 100% ethanol and then xylene to dissolve the alcohol. Next, the tissues were embedded in paraffin and micro-sectioned into 7-µm-thick sections. Slides were deparaffinized with xylene, and 100%, 95%, 80%, and 70% ethanol washed 3 times with PBS, blocked in blocking buffer (PBS, 10% FCS, 10% BSA, 2% triton X100) for 1.5 h at room temperature and incubated with primary antibodies overnight at 4 °C. Following 3 washing steps in PBS, the samples were incubated with secondary antibodies: Donkey anti-rabbit IgG (Invitrogen) or donkey anti-rat IgG (Invitrogen) diluted 1:200 in blocking buffer for 2 h at RT, washed 3 times, stained with DAPI and mounted. As a negative staining control, the primary antibody was omitted and replaced by a blocking buffer. Signals were visualized and digital images were obtained using a Nikon TL microscope for the cross sections and Nikon spinning disk confocal microscope for the wholemount tissues.

## Micro-computed tomography analysis
Maxillae were scanned using a high-resolution scanner (µCT 40, Scanco Medical AG, Bassersdorf, Switzerland). Measurements were taken at an operating voltage of 70 kVp and 114 µA current, with an exposure time of 200 ms and voxel resolution of 12 µm in all three spatial dimensions. To precisely quantify volumetric bone loss, quantitative three-dimensional measurements of teeth were performed. Bone loss analysis for teeth was performed in two methods: as previously reported (Wilensky et al.[64]). In short, the sagittal plane of the specimens was set parallel to the X-ray beam axis. The scans were Gaussian filtered and segmented using a multi-level global thresholding procedure for the segmentation of bone. Residual supportive bone volume around natural teeth was determined separately for the buccomesial and the buccodistal roots. To quantify the residual supportive bone volume around the second molar, an occlusal plane was set between the distal marginal ridge of the third molar and the mesial marginal ridge of the first molar. A reference plane was set at 216 µm below the cementoenamel junction and in parallel to the occlusal plane. After setting the reference plane, the volume of bone that was present coronal to it was measured around the mesiobuccal and distobuccal roots separately. The mesiodistal length of alveolar bone considered during measurement was 120 µm and 144 µm for the mesiobuccal and the distobuccal roots, respectively. The results represented the residual bone 3 mm above the reference plane. In the second method, DICOM files were extracted from the scanner and analyzed in the OnDemand3D Application. The area between the cementoenamel junction and the alveolar bone level was marked and measured on the buccal and palate sides.

## RNA extraction and RT-PCR
For RNA isolation, the excised gingival tissues and tongues were homogenized in 500 µl TRI reagent (Sigma) using an electric homogenizer (IKA labortechnik) and RNA was extracted according to the manufacturer's instructions. cDNA synthesis was performed using the qScript cDNA Synthesis Kit (Quanta-BioSciences). RT-PCR reactions (10 µL volume) were performed using Power SYBR Green PCR Master Mix (Quanta-BioSciences) and specific primers to the examined gene. Sequences of the various primers used in this study are provided in the supplementary data. The following reaction conditions were used: 10 min at 95 °C, 40 cycles of 15 s at 95 °C, and 60 s at 60 °C. The samples were normalized to GAPDH as control mRNA, by change in cycling threshold (ΔCT) method and calculated based on 2-ΔΔCT.

## RNA-Seq differential expression analysis
Raw reads were processed according to the QuantSeq User Guide recommendations (see (1)), reads were trimmed at their 5′ end to remove the first 12 bases, then low-quality and technical bases were removed from the 3′ end using cutadapt (version 1.12) (3). Finally, low quality reads, with more than 30 percent of the bases with quality below 20, were filtered out using the FASTX package (version 0.0.14). Processed reads were aligned against the mouse genome using TopHat (v2.1.1) (4). The genome version was GRCm38, with annotations from Ensembl release 89. Htseq-count (version 0.6.0) (5) was then used for quantification of raw counts per gene per sample, excluding short or otherwise unwanted gene types, such as rRNA or miRNA. Normalization and differential expression analysis were performed with the DESeq2 package (version 1.12.4) (6). Genes with a sum of counts less than 10 over all samples were filtered out prior to normalization. Differential expression, comparing 8-weeks to 1-week-old mice, was calculated with default parameters, except not using the independent filtering algorithm. Statistical significance 6 thresholds were taken as the adjusted p-value (padj) <0.1. Exact commands with the full parameters used can be found under the GEO accession.

## Gene set enrichment analysis
Whole differential expression data (1) were subjected to gene set enrichment analysis using GSEA (7). GSEA uses all differential expression data (cut-off independent) to determine whether a priori–defined sets of genes show statistically significant, concordant differences between two biological states. GSEA was run against the hallmark gene set collection from the molecular signatures database (MSigDB, v6.2, July 2018).

## Microbiota analysis
The oral cavity of anesthetized individual PBS and DT-injected Langerin-DTR mice were swabbed for 30 s and the swabs were then processed for DNA isolation using the MoBio PowerSoil kit according to the manufacturer's instructions. Library preparation, sequencing, and analysis were performed by Hay Laboratories (Israel) as follows: Libraries were prepared using a two-step PCR protocol. In the first step, the V4 region of the 16 rRNA gene was amplified using primers 515F (GTGYCAGCMGCCGCGGTAA) and 807R (GGACTACNVGGGTWTC-TAA) (Earth Microbiome Project). The PCR reactions were cleaned using AMPure beads, and then subjected to a second PCR using the Fluidigm Access Array Primers for Illumina to add the adapter and index sequences. After cleaning the second PCR using AMPure beads, the libraries were quantified by Qubit and the size was determined by Tapestation analysis. The libraries were then sequenced on an Illumina Miseq using the V2 kit for 500 cycles (250 × 2, paired-end reads). Data analysis was performed using the CLC Bio Genomics Workbench and Microbial module referring to the Greengene database. Diversity-related statistical tests were carried out in R using the following packages: veganand phyloseq (McMurdie and Holmes[65]). Taxonomy and OTU tables were input into phyloseq package in R to obtain alpha and beta diversity metrics. The a and b diversities were calculated via Shannon index and Bray-Curtis dissimilarity respectively. To illustrate patterns of bacterial community structure, we performed non-metric multidimensional scaling (NMDS) ordination of Bray-Curtis dissimilarity. Permutational Multivariate Analysis of Variance Using Distance Matrices (PERMANOVA) was used to assess the significance between groups on NMDS.

To examine the absolute abundance of certain bacteria, DNA was prepared from oral swabs using Zymo Quick-DNA Fungal\Bacterial DNA Extraction Kit according to the manufacturer's instructions, and then directly subjected to RT-PCR analysis using designated primers. RT-qPCR reaction was performed using Power SYBR Green PCR Master Mix (Quanta-BioSciences IncTM.). The following reaction conditions were used: 10 min at 95 °C, 40 cycles of 15 s at 95 °C, and 60 s at 60 °C. The samples were normalized to the 18 S as control mRNA, by change in cycling threshold (DCT) method and calculated based on 2-DCT.

## FITC application

In all, 20 mg of FITC (Sigma) was dissolved in 100 μl DMSO (sigma), and the solution was diluted in acetone (1:1). Mice were anesthetized, and 40 μl of the solution was carefully applied to the gingiva. The lymph nodes were excised 2 days after the application for analysis.

## Mechanical damage

Mice were anesthetized, and a swab dipped in sterile PBS was inserted between the right cheek and the upper teeth. The swab was used to rub the gingival tissue going back and forth and then inserted between the right upper teeth and the tongue, followed by rubbing the gingival tissue again in the same manner. This process was then done on the left upper teeth as well. In another experiment, $5 \times 10^5$ CellTrace-labeled splenocytes from CD45.1$^+$ OT-I mice were transferred intravenously to CD45.2$^+$ langerin-DTR mice either pre-injected with DT or PBS. Epithelial damage was induced as described above using a cotton swab soaked with the OVA antigen. The proliferation of the OT-I CD8$^+$ T cells was analyzed 3 days later in the cervical LNs by flow cytometry.

## Preparation of soft diet

Mice were fed with irradiated food (Teklad 2918) as a solid diet. To prepare the soft diet, the exact food was dissolved in distilled water for 3 h in a biological hood and then provided to the mice in a sterile plastic dish on the cage floor. Water was similarly supplied to both groups.

## Statistical analysis

Data are expressed as means ± SEM. Statistical tests were performed using two-sided, unpaired $t$-test comparing two groups, and one-way ANOVA comparing more than two groups using GraphPad Prism. A $P$ value of $<0.05$ was considered significant, Detailed information on the n of biological samples and animals used can be found in figure legends.

## Reporting summary

Further information on research design is available in the Nature Portfolio Reporting Summary linked to this article.

## Data availability

The data that support this study are available in the main text and the supplementary materials and available from the corresponding author upon request. The RNA-seq data generated in this study have been deposited in the NCBI Gene Expression Omnibus GEO database under accession code GSE232790. Source data are provided with this paper.

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

## Acknowledgements

We thank Abed Nasereddin, Idit Shiff, Inbar Plaschkes, Yuval Nevo, and Hadar Benyamini for the RNAseq and bioinformatic analysis. This work was supported by Israel Science Foundation Grant 2272/20 (A.-H.H.).

## Author contributions

Conceptualization: Y.J., and A-H.H. Methodology: Y.J., Y.S, R.N., O.S., K.Z., Y.S., O.B., Y.H., C.N., G.B.C., P.K., O.Y., O.G., and L.E.-B. Visualization: Y.N. Funding acquisition: A.-H.H. and A.W. Contribution of germ-free mice: H.S. and E.E. Writing: A.-H.H.

## Competing interests

The authors declare no competing interests.
