## [Peer Review File · Nature Communications]

Langerhans cells shape postnatal oral homeostasis in a mechanical-force-dependent but microbiota and IL17-independent mannerREVIEWER COMMENTS

Reviewer #1 (Remarks to the Author):

The oral mucosa harbors several immune cell subsets including epithelial Langerhans cells. The authors asked in this study whether LCs are involved in establishing oral immunity after birth. They analyzed critical events during oral immune cell development, at birth, after weaning, i.e. during start of mastication, where shear forces are known to influence LC activation and migration to regional lymph nodes. The involvement of LCs during the development of oral leukocytes and immune mediators remained undefined. As bone regulation is critical for tooth formation, whether/how the oral immune system is involved in this remained elusive. They describe major changes of epithelial and sub-epithelial immune cell composition following weaning. They then studied the effects of key environmental parameter, i.e. the role of microbiota known to seed the epithelium after birth, as well as mastication. Major findings of their study are that LCs are positively regulated by microbiota consistent with their previous findings in adult mice, but LC depletion fails to influence microbiome composition. Moreover, they identified mastication as a key factor required for the development of the oral immune system and for physiologic bone resorption. LCs are key mediators in these processes, as elegantly demonstrated using LC depletion experiments. The authors propose a novel mechanism of bone remodeling during early development that differs from adult mice. This study provides interesting new insights into the roles of microbiota versus shear forces in the development of the oral immune system. The study is technically well conducted, but a little difficult to read. The following points need to be addressed:

Figure 1A nicely shows the changes in the immune cell composition using single cell analyses. As the study focusses on LCs later on, it would be nice to see LCs in these diagrams, or at least to describe why LCs can not be resolved in these diagrams, in context to Figure 1A. Moreover, a major shift from neutrophils to monocytes can be seen in the lamina pro-propria, however this does not translate to the data presented in Figure 1B. This should be clarified.

Figure 2 compares results from germ-free mice vs controls. Interestingly significant changes are noticeable only between weeks 4 to 8. These microbiota-dependent findings in adult animals should be described and discussed in more detail.

Migration cells are elegantly traced using FITC painting. Figure 4F shows that only a minor subset of FITC+ cells express Langerin (22.4% at week 4). What are the FITC+Langerin- cells? Are these LCs that have lost Langerin or are these other DCs? A similar question arises when looking at Figure 5B.

To the opinion of this reviewer, Figure 4 might be optionally moved to supplementary figures.

Ad Introduction lines 75 to 76: is there an error in this sentence with regards to Treg?

Ad Results: as also mentioned above, text description of the single cell data needs to be improved, what are the upper and lower diagrams in Figure 1A, why were these separated, rather than in one single diagram? Some key methods aspect should be mentioned in this Results section for the sake of clarity

line 146: a typo: waning

Discussion: it would be interesting to speculate what microbiota-derived factor(s) might promote LC differentiation? - add a short paragraph on this

Add a paragraph on potential relevance of these findings to human oral immune cell development; are there correlative findings in human that fit to this model; how might findings in mouse regarding bone resorption might translate to human (perinatal and early childhood).

Reviewer #2 (Remarks to the Author):

In this report, Jaber Y et al. demonstrated that in early life, gingival Langerhans cells (LCs) sense and respond to mechanical force as epithelial damage by migrating to the lymph nodes (LNs) to regulate gingival immunity. The authors showed that inducing mechanical damage by brushing the gingiva promoted LC migration to the LNs. Additionally, compared to the soft diet-fed mice, the solid diet-fed mice showed an increased number of adaptive CD4+ T and B cells, indicating the establishment of gingival immunity. Conversely, the conditional ablation of LCs by the administration of diphtheria toxin (DT) to the mice decreased the frequencies of Treg cells and reduced IFN- γ production by the CD4+ T cells. They also found that DT-injected mice showed an impaired gingival epithelial integrity, indicating that LCs can maintain the epithelial integrity in early life. Interestingly, the authors found that the soft diet-fed mice showed less bone loss without IL17 downregulation at 8 weeks after birth compared with that in solid diet-fed mice. In line with this, DT injection inhibited bone loss despite the increase in the number of IL-17A producing CD4+ T cells. Thus, the authors demonstrated that bone loss associated with epithelial integrity during the weaning period is IL-17-independent, unlike that in adult life.

These findings are interesting and have provided new mechanistic insights into gingival immunity in early life. The experimental design is straightforward and reasonably executed. The data analysis and interpretation are appropriate. The overall conclusion is, at least in part, supported by the authors' experimental findings.

I hope the authors would be willing to consider some suggestions to improve the clarity and potential clinical significance of their findings.

1. The key point of this study is the regulation of postnatal gingival immunity by the LCs. Nevertheless, how the LCs regulate T-cell activation is still unclear. The authors claim that the LCs migrate to the LNs due to mechanical force-activated T cells. Although the authors demonstrated that LC ablation reduced several immunological pathways, these data are not sufficient to support the scenario of T-cell activation by the LCs in the LN during early life. Therefore, the interactions between the LCs and T cells in the LNs, such as the presenting of exogenous antigens, should be shown.
2. Although the authors have demonstrated that mechanical damage by brushing the gingiva promoted LC migration to the LNs, the role of LCs in gingival immunity under mechanical force was mainly investigated using solid and soft diets. Therefore, the number of LCs in the LNs of the mice in both the solid and soft diet groups should be investigated to confirm that the solid diet induced LC migration to the LNs.
3. Interestingly, the study showed that ablation of LCs by DT injection resulted in substantial penetration of the FITC dye into the epithelium. Perhaps, equally interesting to observe would be the differences in the HE-stained sections or tight junction protein expression between the LC injection and control groups.
4. The markers that were used for the identification of ILCs need to be mentioned in the manuscript.
5. In the figures, including Fig. 2A–B and Fig. 6F, the authors should note at how many weeks after birth each data point was taken.
6. The manuscript needs to be carefully reviewed as several mistakes can be spotted. For instance, "Fig. 5D" in line 155 and "Fig. 5E" in line 157 might be "Fig. 3D" and "Fig. 3E," respectively. Similarly, the sentence "higher numbers of TRAP+ cells were detected in gingival cross-sections of mice fed with a soft than solid diet" in lines 210–211 is not consistent with Fig. 5I.

Reviewer #3 (Remarks to the Author):

The manuscript from Jaber et al. provides an intriguing story that highlights the importance of Langerhans cells, microbiota and masticatory forces in shaping the maturation of gingival immunity. The authors use multiple modalities and approaches to generate data that is novel with findings that build upon current knowledge in the field. The manuscript could, however, benefit from additional experimentation and data inclusion to further support the author's conclusions.

Major Comments:

- 1) Flow cytometry is used as a key readout to track immune cells over time and under different conditions. Although the populations measured are extensive, all the data is represented in % of total (in case of CD45 cells) or of CD45. Inclusion of total numbers would be equally important to ensure that population changes shown as % are not dictated by a big elevation or decrease of another population. Even representation of total CD45 number quantification would help define these changes better.
- 2) The impact of the microbiota on gingival immunity is considerably downplayed in the manuscript. The authors suggest that the microbiota plays simply an early role and then mastication takes on a more important role in gingival immunity. The data shows that there is differential mediator/TF expression in figure 2C between SPF and GF only at the 8-week timepoint. In addition, the GF condition contributes to altered immune cell presence locally including LCs and gamma/delta T cells that are noted as important regulators of the local environment. The LC quantification is completed at 4 weeks and the measurement of the other immune cells in Figure 2 is presumably also 4 weeks (although not indicated). Without SPF vs GF quantification at 8 weeks to show that the levels of these immune cells return to SPF levels, it is hard to make the conclusion that the microbiota has limited effects past weaning.
- 3) Mastication is proposed to be a key trigger for LC control of the local environment presumably through increasing LC trafficking to the LN. The authors provide experimentation comparing the solid vs soft diets showing that mastication plays an important role in regulating immune cell populations and bone resorption. However, for LC trafficking, a direct damage approach is used instead. Since the soft diet is used to make key conclusions, it would be informative to continue the diet comparison with FITC trafficking assay. In addition, can the authors explain why the percentage of LCs is much lower with the solid diet in Figure 5 as compared to Figure 3a. Quantification of LCs by IHC as done in Figure 3 and LC number from these flow cytometry experiments would help to further support LC differences between the two diet conditions.
- 4) The authors show compelling data that the depletion of LC contributes to changes in the local gingival immunity. Is this regulation due to its general presence in the epithelium or dependent on its trafficking to the lymph node? Can trafficking be blocked long-term (post-weaning) to distinguish these effects on the local immunity? This would be insightful considering the general narrative indicating that masticatory forces drives LC trafficking to influence local immunity.
- 5) Figures should be improved to facilitate reader interpretation of the data.
 - a) The panel organization in Figure 4, 6, and 7 are sporadic and difficult to follow.
 - b) Timepoint labeling (e.g. 0, 4, and 8 weeks) for most experiments were included in the majority of panels, while missed for others. Adding them would make things more consistent and the data more interpretable particularly since the maturation state is important to the story.
 - c) In Figure 1, are the bottom t-SNE panels only of the CD45+ cells? If yes this should be labelled or some relation between the CD45+ cells in the upper panels should be made.

Point-by-point response

Reviewer #1:

The oral mucosa harbors several immune cell subsets including epithelial Langerhans cells. The authors asked in this study whether LCs are involved in establishing oral immunity after birth. They analyzed critical events during oral immune cell development, at birth, after weaning, i.e. during start of mastication, where shear forces are known to influence LC activation and migration to regional lymph nodes. The involvement of LCs during the development of oral leukocytes and immune mediators remained undefined. As bone regulation is critical for tooth formation, whether/how the oral immune system is involved in this remained elusive. They describe major changes of epithelial and sub-epithelial immune cell composition following weaning. They then studied the effects of key environmental parameter, i.e. the role of microbiota known to seed the epithelium after birth, as well as mastication. Major findings of their study are that LCs are positively regulated by microbiota consistent with their previous findings in adult mice, but LC depletion fails to influence microbiome composition. Moreover, they identified mastication as a key factor required for the development of the oral immune system and for physiologic bone resorption. LCs are key mediators in these processes, as elegantly demonstrated using LC depletion experiments. The authors propose a novel mechanism of bone remodeling during early development that differs from adult mice. This study provides interesting new insights into the roles of microbiota versus shear forces in the development of the oral immune system. The study is technically well conducted, but a little difficult to read. The following points need to be addressed:

Response- We thank the reviewer for reading our manuscript carefully and for the positive review. Please find below our responses to the specific comments.

Figure 1A nicely shows the changes in the immune cell composition using single cell analyses. As the study focusses on LCs later on, it would be nice to see LCs in these diagrams, or at least to describe why LCs can not be resolved in these diagrams, in context to Figure 1A. Moreover, a major shift from neutrophils to monocytes can be seen in the lamina pro-propri, however this does not translate to the data presented in Figure 1B. This should be clarified.

Response – As the reviewer mentioned, LCs were not indicated in Fig. 1A since we didn't include the relevant antibodies in the large panel that we used to identify the major leukocytes in the developing gingiva. Nevertheless, this information is provided in Fig. 3, in which we analyzed, in greater depth, the development of gingival LC after birth. Regarding the comment on the neutrophils and monocytes in the lamina propria, Fig. 1B shows the relative percentages of these cells from the CD45⁺ leukocytes. Our analysis revealed that similar to the epithelium, the frequencies of neutrophils and monocytes are reduced during the weaning period as adaptive immunity develops in the gingiva. The graphs in Fig. 1B adequately report this shift.

Figure 2 compares results from germ-free mice vs controls. Interestingly significant changes are noticeable only between weeks 4 to 8. These microbiota-dependent findings in adult animals should be described and discussed in more detail.

Response –According to this recommendation, we discussed in more detail the microbiota-dependent changes in the expression of the various genes described in Fig. 2C (Page 15 lines 23, and Page 16 lines 1-4).

Migration cells are elegantly traced using FITC painting. Figure 4F shows that only a minor subset of FITC+ cells express Langerin (22.4% at week 4). What are the FITC+Langerin- cells? Are these LCs that have lost Langerin or are these other DCs? A similar question arises when looking at Figure 5B.

Response – We fully agree with this important note. Indeed, besides FITC+ LCs, other FITC+ APCs migrate to the lymph nodes during weaning or upon the induction of epithelial damage. Our analysis shows that some of these APCs express EpCAM, which likely represents LCs that have lost langerin expression. The other EpCAM^{neg}Langerin^{neg} APCs might represent either newly developing LCs (these cells are also present in the epithelium) or DC located in the lamina propria. This suggests that various APC subsets respond to the noted immunological challenges, a common behavior of LCs/DCs also observed in other systems. Nevertheless, our LC depletion experiments prove that migrating LCs play an important and non-redundant role in shaping T cell immunity during weaning. To meet the reviewer's comment, we included the above explanation in the revised manuscript (Page 8 lines 12-16).

To the opinion of this reviewer, Figure 4 might be optionally moved to supplementary figures.

Response – We appreciate this comment. However, we believe that the inability of LCs to regulate the oral microbiota during weaning is an important point in our study, and by keeping Figure 4 in the main manuscript, this issue will receive the reader's attention.

Ad Introduction lines 75 to 76: is there an error in this sentence with regards to Treg?

Response- We thank the referee for spotting this error. We rewrote the sentence to clarify that depletion of LCs during periodontitis increases bone loss due to reduced generation of Treg cells (Page 4 line 13).

Ad Results: as also mentioned above, text description of the single cell data needs to improved, what are the upper and lower diagrams in Figure 1A, why were these separated, rather than in one single diagram? Some key methods aspect should be mentioned in this Results section for the sake of clarity

Response- As requested, the text description of Fig. 1A was improved for better clarity (Page 5 lines 7-11).

line 146: a typo: waning

Response- Corrected.

Discussion: it would be interesting to speculate what microbiota-derived factor(s) might promote LC differentiation? - add a short paragraph on this

Response- As recommended, a paragraph discussing the possible involvement of microbiota-derived factors in LC differentiation was added to the revised manuscript (Page 16 lines 14-22).

Add a paragraph on potential relevance of these findings to human oral immune cell development; are there correlative findings in human that fit to this model; how might findings in mouse regarding bone resorption might translate to human (perinatal and early childhood).

Response- As recommended, a paragraph discussing the relevance of our findings to humans was added to the revised manuscript (Page 17 lines 13-21).

Reviewer #2:

In this report, Jaber Y et al. demonstrated that in early life, gingival Langerhans cells (LCs) sense and respond to mechanical force as epithelial damage by migrating to the lymph nodes (LNs) to regulate gingival immunity. The authors showed that inducing mechanical damage by brushing the gingiva promoted LC migration to the LNs. Additionally, compared to the soft diet-fed mice, the solid diet-fed mice showed an increased number of adaptive CD4+ T and B cells, indicating the establishment of gingival immunity. Conversely, the conditional ablation of LCs by the administration of diphtheria toxin (DT) to the mice decreased the frequencies of Treg cells and reduced IFN- γ production by the CD4+ T cells. They also found that DT-injected mice showed an impaired gingival epithelial integrity, indicating that LCs can maintain the epithelial integrity in early life. Interestingly, the authors found that the soft diet-fed mice showed less bone loss without IL17 downregulation at 8 weeks after birth compared with that in solid diet-fed mice. In line with this, DT injection inhibited bone loss despite the increase in the number of IL-17A producing CD4+ T cells. Thus, the authors demonstrated that bone loss associated with epithelial integrity during the weaning period is IL-17-independent, unlike that in adult life.

These findings are interesting and have provided new mechanistic insights into gingival immunity in early life. The experimental design is straightforward and reasonably executed. The data analysis and interpretation are appropriate. The overall conclusion is, at least in part, supported by the authors' experimental findings.

I hope the authors would be willing to consider some suggestions to improve the clarity and potential clinical significance of their findings.

1. The key point of this study is the regulation of postnatal gingival immunity by the LCs. Nevertheless, how the LCs regulate T-cell activation is still unclear. The authors claim that the LCs migrate to the LNs due to mechanical force-activated T cells. Although the authors demonstrated that LC ablation reduced several immunological pathways, these data are not sufficient to support the scenario of T-cell activation by the LCs in the LN during early life. Therefore, the interactions between the LCs and T cells in the LNs, such as the presenting of exogenous antigens, should be shown.

Response- To demonstrate the interactions between LCs and T cells we added a new data set to the revised manuscript (Figure S3 and relevant text on page 10 lines 10-15 and Page 25 lines 11-16). In this experiment, CellTrace-labeled splenocytes from OT-I mice were adoptively transferred to langerin-DTR mice, and epithelial damage was induced in the oral mucosa by rubbing a cotton swab soaked with the

exogenous antigen OVA. Our analysis revealed that depletion of LCs reduced OVA-specific T-cell proliferation in the cervical lymph nodes, indicating that LCs present antigens to T cells upon tissue damage.

2. Although the authors have demonstrated that mechanical damage by brushing the gingiva promoted LC migration to the LNs, the role of LCs in gingival immunity under mechanical force was mainly investigated using solid and soft diets. Therefore, the number of LCs in the LNs of the mice in both the solid and soft diet groups should be investigated to confirm that the solid diet induced LC migration to the LNs.

Response- As requested by the referee, we enumerated LCs in the LNs of mice fed with solid and soft diets. As depicted in the revised Fig. S2, higher numbers of LCs were detected in the cervical LNs of mice provided with a hard diet. Moreover, we examined LC migration from the gingiva to the LNs by painting the gingiva with FITC solution and found higher migratory LCs in mice fed with a high diet (Fig. S2 and the relevant text on page 10 lines 1-3).

3. Interestingly, the study showed that ablation of LCs by DT injection resulted in substantial penetration of the FITC dye into the epithelium. Perhaps, equally interesting to observe would be the differences in the HE-stained sections or tight junction protein expression between the LC injection and control groups.

Response- As requested by the referee, we provided in the revised manuscript H&E-stained gingival cross-sections and immunofluorescence staining against Claudin 4 and ZO1. No significant differences were found between LC-depleted mice to the control, suggesting that the higher permeability cannot be explained by the expression of these genes. This data is presented in Fig. S4 of the revised manuscript (page 12 lines 20-23).

4. The markers that were used for the identification of ILCs need to be mentioned in the manuscript.

Response- ILCs were identified as the following: CD45⁺CD90⁺MHCII^{neg}CD11b^{neg}γδTCR^{neg}αβTCR^{neg} (in some experiments, instead of αβTCR, the CD4 and CD8 markers were used), and further characterization was based on either CD103 or CD69 expression. This information is now included in the revised Fig. S1.

5. In the figures, including Fig. 2A–B and Fig. 6F, the authors should note at how many weeks after birth each data point was taken.

Response- According to this comment, we clarified in the figure and figure legend that 8 weeks-old mice were analyzed (Page 33 line 17, and page 35 line 8).

6. The manuscript needs to be carefully reviewed as several mistakes can be spotted. For instance, “Fig. 5D” in line 155 and “Fig. 5E” in line 157 might be “Fig. 3D” and “Fig. 3E,” respectively. Similarly, the sentence “higher numbers of TRAP+ cells were detected in gingival cross-sections of mice fed with a soft than solid diet” in lines 210–211 is not consistent with Fig. 5I.

Response- Thank you for noticing these errors, which are now corrected, in addition, we carefully reviewed the manuscripts to avoid such mistakes.

Reviewer #3:

The manuscript from Jaber et al. provides an intriguing story that highlights the importance of Langerhans cells, microbiota and masticatory forces in shaping the maturation of gingival immunity. The authors use multiple modalities and approaches to generate data that is novel with findings that build upon current knowledge in the field. The manuscript could, however, benefit from additional experimentation and data inclusion to further support the author's conclusions.

Major Comments:

1)Flow cytometry is used as a key readout to track immune cells over time and under different conditions. Although the populations measured are extensive, all the data is represented in % of total (in case of CD45 cells) or of CD45. Inclusion of total numbers would be equally important to ensure that population changes shown as % are not dictated by a big elevation or decrease of another population. Even representation of total CD45 number quantification would help define these changes better.

Response- As recommended, we added to Fig.1 the total numbers of CD45+ cells in the tissues at each time point.

2)The impact of the microbiota on gingival immunity is considerably downplayed in the manuscript. The authors suggest that the microbiota plays simply an early role and then mastication takes on a more important role in gingival immunity. The data shows that there is differential mediator/TF expression in figure 2C between SPF and GF only at the 8-week timepoint. In addition, the GF condition contributes to altered immune cell presence locally including LCs and gamma/delta T cells that are noted as important regulators of the local environment. The LC quantification is completed at 4 weeks and the measurement of the other immune cells in Figure 2 is presumably also 4 weeks (although not indicated). Without SPF vs GF quantification at 8 weeks to show that the levels of these immune cells return to SPF levels, it is hard to make the conclusion that the microbiota has limited effects past weaning.

Response- We thank the referee for highlighting this point. Please note that in Figure 2 we compared 8 weeks-old GF and SPF mice (we clarified this in the revised manuscript). Therefore, as mentioned by the referee, it can be concluded that the microbiota has limited effects after weaning.

3)Mastication is proposed to be a key trigger for LC control of the local environment presumably through increasing LC trafficking to the LN. The authors provide experimentation comparing the solid vs soft diets showing that mastication plays an important role in regulating immune cell populations and bone resorption. However, for LC trafficking, a direct damage approach is used instead. Since the soft diet is used to make key conclusions, it would be informative to continue the diet comparison with FITC trafficking

assay. In addition, can the authors explain why the percentage of LCs is much lower with the solid diet in Figure 5 as compared to Figure 3a. Quantification of LCs by IHC as done in Figure 3 and LC number from these flow cytometry experiments would help to further support LC differences between the two diet conditions.

Response- As requested by the referee, we quantified the migration of LCs to LNs of mice with a solid or soft diet. As depicted in Fig. S2, reduced numbers of FITC+ LCs (and other APCs) were found in LNs of mice fed a soft diet compared to a hard diet. This reinforces our previous data using direct damage, suggesting that LC migration is also controlled by indirect damage caused by masticatory forces. As for the percentages of LCs in Figure 5 and Figure 3a, it should be mentioned that the percentages of APCs and LCs (as well as other leukocytes) are affected by the isolation procedure, which varies between experiments conducted at different times. This is even more relevant when processing cells from epithelial tissues that require multiple processing steps (ie, enzymatic and mechanical digestion). However, this is not a problem in our experiment, as the control and test groups were processed and analyzed side by side. Regarding the latter part of this comment, as recommended, we quantified LC numbers in epithelial sheets collected from mice fed soft and solid diets. As depicted in the revised Figure 3C, the gingiva of mice fed a soft diet contains higher numbers of LCs.

4)The authors show compelling data that the depletion of LC contributes to changes in the local gingival immunity. Is this regulation due to its general presence in the epithelium or dependent on its trafficking to the lymph node? Can trafficking be blocked long-term (post-weaning) to distinguish these effects on the local immunity? This would be insightful considering the general narrative indicating that masticatory forces drives LC trafficking to influence local immunity.

Response- We thank the referee for this comment. However, we are not aware of a method by which LC migration can be specifically blocked without blocking the migration of other DCs. (In fact, we are not aware of a method by which DC/LC migration can be blocked for an extended period of time without damaging the tissue). As our data show, different DC subsets migrate out of the gingiva due to chewing/mechanical damage, thus it would be impossible to isolate the exclusive role of LCs. Therefore, we believe that our LC depletion strategy (including a new dataset added to the revised manuscript) provides compelling evidence for the ability of LCs to control gingival immunity during weaning.

5)Figures should be improved to facilitate reader interpretation of the data.

Response- We improved the figure for better clarity, we hope the referee will find them clearer.

a)The panel organization in Figure 4, 6, and 7 are sporadic and difficult to follow.

Response- The figures were adjusted to improve clarity.

b)Timepoint labeling (e.g. 0, 4, and 8 weeks) for most experiments were included in the majority of panels, while missed for others. Adding them would make things more consistent and the data more interpretable particularly since the maturation state is important to the story.

Response- As recommended, we improved the timepoint labeling in the figure of the revised manuscript.

c) In Figure 1, are the bottom t-SNE panels only of the CD45+ cells? If yes this should be labelled or some relation between the CD45+ cells in the upper panels should be made.

Response- As recommended, we labeled the lower panel in Fig. 1B as CD45+ cells.

REVIEWERS' COMMENTS

Reviewer #2 (Remarks to the Author):

I am happy that the manuscript was improved after revision. The authors have addressed my concerns and comments.

Reviewer #3 (Remarks to the Author):

The authors have made numerous changes they have greatly improved the quality of the work and further supports the conclusions presented. They have addressed most comments raised by this reviewer, however, there are still a few small points that could be addressed to further improve the work.

1. Related to previous comments 1 and 5C, the authors have added CD45 cell numbers and have added additional labeling and description in the text to define the t-SNE plots presented in Figure 1. Since CD45+ cells increase over time, reductions in innate leukocytes should be referred as changes in the proportions of these cells and not levels as currently used in the text. Based on the increase in total immune cells with a decrease in proportion, the levels of these cells are likely stable.

To simplify the presentation of this data, the authors should consider moving the bottom panels t-SNE plots (CD45+ cells) to panel B along with the quantification of these populations.

2. Related to previous comment 3, the authors nicely show the differences in migratory capacity between the hard and soft diets which furthers supports their claims. Flipping this new data into the main figure and placing the direct damage model (currently Fig 5E) to supplementary would make for a more cohesive figure.

3. Related to comment 5a), the organization of panels has been improved but could still be optimized further so that panels read from left to right and top to bottom.

Point-by-point response

Reviewer #2 (Remarks to the Author):

I am happy that the manuscript was improved after revision. The authors have addressed my concerns and comments.

Response – We thank reviewers #1 and #2 for accepting our revisions.

Reviewer #3 (Remarks to the Author):

The authors have made numerous changes they have greatly improved the quality of the work and further supports the conclusions presented. They have addressed most comments raised by this reviewer, however, there are still a few small points that could be addressed to further improve the work.

1. Related to previous comments 1 and 5C, the authors have added CD45 cell numbers and have added additional labeling and description in the text to define the t-SNE plots presented in Figure 1. Since CD45+ cells increase over time, reductions in innate leukocytes should be referred as changes in the proportions of these cells and not levels as currently used in the text. Based on the increase in total immune cells with a decrease in proportion, the levels of these cells are likely stable.

Response – As suggested, the word “levels” was replaced with “proportions”.

To simplify the presentation of this data, the authors should consider moving the bottom panels t-SNE plots (CD45+ cells) to panel B along with the quantification of these populations.

Response – Figure 1 was revised as suggested.

2. Related to previous comment 3, the authors nicely show the differences in migratory capacity between the hard and soft diets which furthers supports their claims. Flipping this new data into the main figure and placing the direct damage model (currently Fig 5E) to supplementary would make for a more cohesive figure.

Response – As recommended, we transferred the damage model from Fig. 5 to the supplementary data and included the new data in the main figure. In addition, to comply with the editorial request of a 350-word limitation in the figure legends, we split figure 5 into 2 separate figures, therefore, the revised manuscript contains 8 figures (instead of 7 in the original version).

3. Related to comment 5a), the organization of panels has been improved but could still be optimized further so that panels read from left to right and top to bottom.

Response – The panels were rearranged as much as we can according to this comment.